# AGENT LEARNING VIA EARLY EXPERIENCE

## ABSTRACT

A long-term goal of language agents is to learn and improve through their own experience. However, training agents from experience data with reinforcement learning remains difficult in many environments, which either lack verifiable rewards (*e.g.*, websites) or require inefficient long-horizon rollouts (*e.g.*, multi-turn tool use). As a result, most current agents rely on supervised fine-tuning on expert data, which is difficult to scale and generalizes poorly. This limitation stems from the nature of expert demonstrations: they capture only a narrow range of scenarios, and expose the agent to limited environment diversity. We address this limitation with a middle-ground paradigm we call *early experience: interaction data generated by the agent's own actions, where the resulting future states serve as supervision without reward signals*. Within this paradigm we study two strategies of using such data: (1) Implicit world modeling, which uses collected states to ground the policy in environment dynamics; and (2) Self-reflection, where the agent learns from its suboptimal actions to improve reasoning and decision making. We evaluate across eight diverse environments and multiple model families. Our approaches consistently improve effectiveness and out-of-domain generalization, highlighting the value of early experience. Moreover, in environments with verifiable rewards, our results provide promising signals that early experience offers a strong foundation for subsequent reinforcement learning, positioning it as a practical bridge between imitation learning and fully experience-driven agents.

## 1 INTRODUCTION

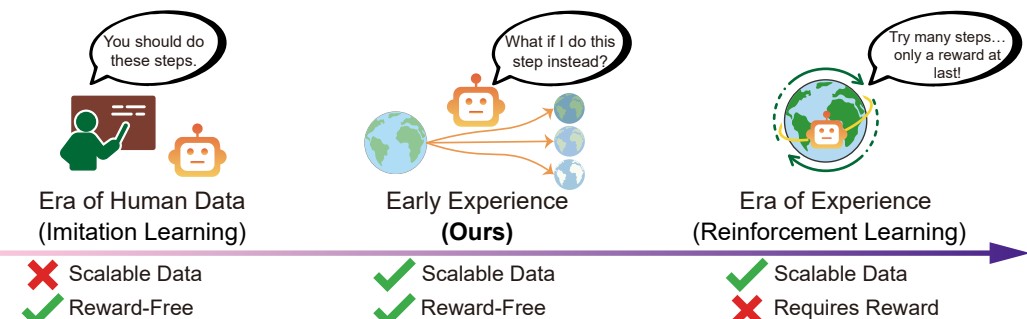

Figure 1: Progression of training paradigms for language agents. **Left**: The *Era of Human Data* relies on expert demonstrations, where supervision comes from human-/expert-curated actions; it is reward-free (*i.e.*, does not require the environment to provide verifiable reward) but not data-scalable. **Right**: The envisioned *Era of Experience* builds upon environments with verifiable rewards, using them as the primary supervision for reinforcement learning; however, many environments either lack such rewards (Xue et al., 2025) or require inefficient long-horizon rollouts (Xie et al., 2024a). **Center**: Our *Early Experience* paradigm enables agents to propose actions and collect the resulting future states, using them as a scalable and reward-free source of supervision.

Autonomous agents (Russell & Norvig, 1995; Franklin & Graesser, 1997) have long been a central goal of artificial intelligence, aiming to perceive, act, and learn in complex environments to accomplish goals without human intervention. This vision is becoming increasingly realistic with the emergence of language agents (Su et al., 2024; Sumers et al., 2024), which are built on top of large language models (LLMs; OpenAI (2024)). Powered by knowledge obtained from large-scale pretraining and

the flexibility of the language interface, language agents are now being applied across a wide range of environments. They can navigate websites and mobile applications (Zheng et al., 2024a; Deng et al., 2023; Zhou et al., 2024; Trivedi et al., 2024), control diverse tools (Xie et al., 2024a; Gu et al., 2024), and assist in scientific research (Chen et al., 2025; Lou et al., 2025), showing strong potential as a foundation for the next generation of intelligent systems.

To build such language agents, one promising solution is reinforcement learning (RL), where agents are trained by optimizing for long-term rewards returned by the environment. This paradigm has enabled traditional agents such as AlphaGo (Silver et al., 2016) to achieve superhuman performance in domains with well-defined environments and reward structures, such as Atari games (Bellemare et al., 2013) and the game of Go, echoing the vision of an emerging *era of experience* (Silver & Sutton, 2025) for language agents. However, applying RL to real-world language agents remains highly challenging now. Many environments of interest lack verifiable or dense reward signals, especially in open-ended settings such as websites where platforms do not expose ground truth feedback. For example, a form may appear to be submitted successfully, but the agent receives no indication of whether each piece of information was filled out correctly. In addition, tasks in multi-turn tool-use environments often involve long interaction sequences (Xie et al., 2024a; Jin et al., 2025) with delayed or ambiguous outcomes, making credit assignment and training inefficient and unstable.

As a workaround, most current language agents are instead trained on expert-curated data with supervised fine-tuning (SFT; Deng et al. (2023); Pahuja et al. (2025); Prabhakar et al. (2025)). This paradigm bypasses the need for reward signals by learning from human demonstrations, where agents map states to actions using static datasets. While SFT is straightforward and efficient to train, it has inherent limitations. The agent under this paradigm does not interact with the environment during training; it does not observe the outcomes of its own actions. This restricts its ability to learn from failure, refine its decision making, or generalize to unseen situations (Chu et al., 2025). Furthermore, this approach assumes the data are expert or near-optimal, yet scaling high-quality human demonstrations is difficult. More critically, it locks the agent into a passive role, bound by the imagination and coverage of its training data rather than actively learning from its own experience. Given these limitations and that reliable reward signals are often unavailable aforementioned, *how can we train agents to grow from its own experience, **without any external reward signals**?*

Motivated by these limitations, we introduce the *early experience* paradigm, a middle ground between imitation learning and reinforcement learning, as shown in Figure 1. In this setting, agents learn not only from human-curated data but also from future states driven by their own proposed actions in the environment. These future states are the agent's own experience, and can be transformed into supervision signals that enable it to grow directly from the consequences of its actions without relying on external reward signals. We explore two strategies to transform these future states as supervision: (1) **Implicit World Modeling**: using the collected future states to help the agent build internal representations of environment dynamics, allowing it to better understand the environment by predicting the future states. (2) **Self-Reflection**: guiding the agent to compare its behavior with expert demonstrations, identify suboptimal decisions, and extract lessons to improve future decision making. Both strategies share the same principle: in the absence of external rewards, the agent's own actions and the resulting future states can still constitute experience that serve as a direct source of supervision. By turning future states generated from its own actions into learning signals, the language agent can continually improve without relying on additional human data or external rewards.

We comprehensively evaluate early experience across eight diverse environments, spanning embodied navigation, web navigation, multi-turn tool-use, long-horizon planning, and multi-domain API tasks, using multiple base architectures. Across all settings, both methods consistently outperform purely imitation learning baselines, with average absolute gains of **+9.6** in success rate and **+9.4** in out-of-domain generalization. Moreover, in environments where verifiable rewards are available, initializing RL with checkpoints trained with early experience methods leads to substantially stronger performance compared to standard imitation-learning warm starts, improving final success rates by up to **+6.4**. This shows that the performance gain from early experience stage can carry over to the final model's performance after RL. Beyond these empirical gains, our analysis shows that early experience enables capabilities unattainable through imitation learning alone. It scales effectively, achieving comparable or superior performance with only half or even less of the expert data. The paradigm applies seamlessly to larger models, preserving its effectiveness across scales. These results show that early experience is not merely an alternative to imitation learning, but a practical

and scalable bridge to reinforcement learning, delivering both immediate gains in effectiveness and long-term benefits for *era of experience* training regimes.

Our contributions are summarized as follows: **(1)** We advocate and formalize the *early experience* paradigm as a practical and scalable bridge between imitation learning and reinforcement learning for building autonomous language agents. It empowers agents to convert their own experience into learning signals without relying on external rewards and can be seamlessly integrated into existing training pipelines. **(2)** We propose and systematically study two training strategies under this paradigm: implicit world modeling, which enhances decision making by modeling environment dynamics directly from collected experience, and self-reflection, which distills fine-grained lessons from the agent's own actions. **(3)** We conduct a comprehensive evaluation across eight diverse environments and multiple model families. Our methods consistently improve task effectiveness, out-of-domain generalization, and downstream reinforcement learning performance, achieving state-of-the-art results on several benchmarks and offering actionable insights through detailed analysis.

## 2 RELATED WORK

We discuss training paradigms for language agents in this section, and we discuss more related work on supervision from exploration in Appendix C.

**Supervised Fine Tuning (SFT).** Most language agents (Yao et al., 2022; Deng et al., 2023; Hong et al., 2024; Furuta et al., 2024; Pahuja et al., 2025) are trained with SFT, also known as imitation learning or behavior cloning in the RL literature, on expert trajectories, especially in complex settings such as the web (Zhou et al., 2024) or operating systems (Xie et al., 2024b). These trajectories may be human-annotated (Yao et al., 2022; Deng et al., 2023) or synthesized by stronger language models that follow carefully human-designed workflows (Murty et al., 2024; Pahuja et al., 2025). Although synthetic demonstrations increase coverage, they offer only incremental gains because the underlying supervision signal is still static. SFT thus provides dense, reward-free supervision signals but remains limited by the cost of high-quality demonstrations (Qi et al., 2025) and leaves agents brittle when they confront novel states (Chu et al., 2025; Deng et al., 2023).

**Reinforcement Learning (RL).** RL trains agents through trial and error, optimising for long-term rewards (Sutton et al., 1998). Although it has achieved impressive results in control, board games, and Atari (Mnih et al., 2013; Silver et al., 2016; Hafner et al., 2020; Schrittwieser et al., 2020), RL remains difficult to apply effectively in language-agent settings (Wang et al., 2025; Qi et al., 2025; Wei et al., 2025a; Feng et al., 2025; Zhou et al., 2025b; Jin et al., 2025; Zhou et al., 2025a). Current studies are still exploratory: many rely on approximate rewards produced by larger teacher models (Qi et al., 2025; Zhou et al., 2025b), or on carefully curated reward functions (Qian et al., 2025) and hand-tuned training recipes (Jin et al., 2025) to maintain stability. The supporting infrastructure is also underdeveloped; most real-world language agent environments lack reliable simulators, standard reset mechanisms, and scalable evaluation platforms (Wang et al., 2025; Feng et al., 2025), making large-scale RL training for language agents costly and brittle. Together, these limitations suggest that scalable RL for language agents is not yet mature, motivating a paradigm that bridges current imitation-based training and future fully experience-driven learning (RL).

## 3 PRELIMINARIES

We formalize the language agent decision-making problem as a Markov Decision Process (MDP; Bellman (1957)), which provides the mathematical foundation for our early experience paradigm.

We consider an MDP defined by the tuple $\mathcal{M} = (\mathcal{S}, \mathcal{A}, T, R, \gamma, \rho_0)$, where $\mathcal{S}$ denotes the state space and $\mathcal{A}$ represents the action space. The transition function $T \colon \mathcal{S} \times \mathcal{A} \to \Delta(\mathcal{S})$ governs state dynamics, where $\Delta(\mathcal{S})$ denotes the probability simplex over $\mathcal{S}$. The reward function $R \colon \mathcal{S} \times \mathcal{A} \to \mathbb{R}$ provides feedback signals when available, though in many real-world settings this function may be unknown or unverifiable during training. $\gamma \in [0, 1]$ is the discount factor, and $\rho_0 \in \Delta(\mathcal{S})$ specifies the initial state distribution. In language agent environments, states $s \in \mathcal{S}$ encode the environment configuration accessible to the agent, such as webpage contents, tool outputs, or textual environment descriptions. Actions $a \in \mathcal{A}$ correspond to discrete choices such as clicking elements, invoking tools, or generating text responses. The agent maintains a policy $\pi_\theta \colon \mathcal{S} \to \Delta(\mathcal{A})$, parameterized by $\theta$, which maps states to action distributions (Williams, 1992).

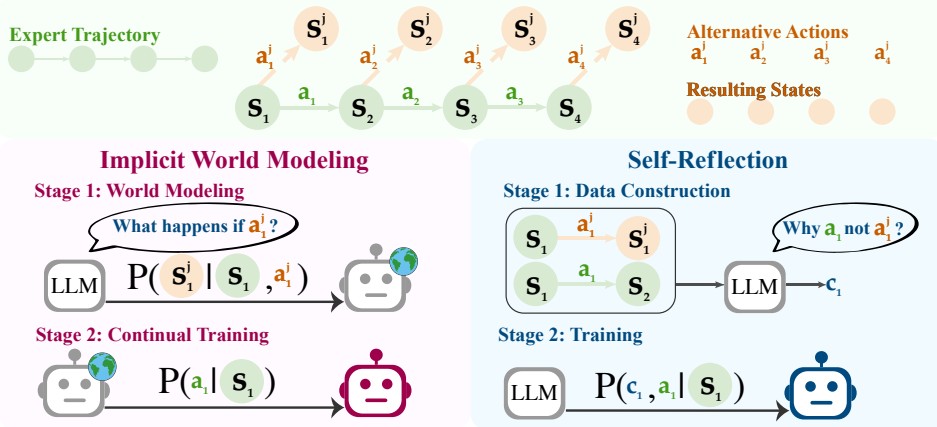

Figure 2: Overview of the two early experience approaches. Implicit world modeling (left) augments expert trajectories with alternative actions and predicted next states, training the policy to internalize transition dynamics before deployment. Self-reflection (right) augments expert actions with self-generated explanations $c_1$, training the policy to reason about and revise its own decisions. Both methods use alternative actions proposed by the initial policy (LLM). The number of alternatives ($K$) is a hyperparameter; for brevity, only one is illustrated.

### 3.1 LEARNING WITHOUT REWARDS

A key challenge in real-world language agent environments is the absence of reliable reward signals. Many environments either lack verifiable rewards entirely or provide only sparse, delayed feedback after long interaction sequences. This motivates learning from alternative supervision sources.

Given a dataset of expert demonstrations $\mathcal{D}_{\text{expert}} = \{(s_i, a_i)\}_{i=1}^{N}$, where $a_i$ denotes the expert action at state $s_i$, imitation learning (Pomerleau, 1991; Schaal, 1996; Hussein et al., 2017) aims to minimize the supervised learning loss:

$$\mathcal{L}_{\text{IL}}(\theta) = -\sum_{i=1}^{N} \log \pi_\theta(a_i \mid s_i). \tag{1}$$

However, this approach suffers from distribution shift and lacks awareness of action consequences. Distribution shift occurs because the agent's learned policy $\pi_\theta$ inevitably deviates from the expert policy during deployment, leading to states not covered in training data where errors compound (Ross et al., 2011). The agent lacks awareness of action consequences because it never observes what happens when it takes non-expert actions; it only sees expert state-action pairs without experiencing the outcomes of alternative choices. This limits its ability to recover from errors or reason about why certain actions fail (Ross & Bagnell, 2010).

## 4 EARLY EXPERIENCE

We introduce the *early experience* paradigm, where language agents improve through interaction with the environment using reward-free but informative future states. To build intuition, consider a language agent learning to book flights on the web. In traditional imitation learning, it only sees expert demonstrations of successful bookings. With early experience, the agent also explores what happens when it clicks different buttons or fills forms incorrectly, observing error messages, page changes, and other outcomes. These observations become learning signals without explicit rewards. Starting from expert trajectories, the agent proposes its own actions at each visited state to collect additional environment feedback through exploration (Thrun, 1992).

### 4.1 NOTATION FOR EARLY EXPERIENCE

For each expert state $s_i$ in the dataset $\mathcal{D}_{\text{expert}} = \{(s_i, a_i)\}_{i=1}^{N}$, we define a candidate action set $\mathcal{A}_i = \{a_i^1, a_i^2, \ldots, a_i^K\}$, where we sample $K$ alternative actions from the initial policy $\pi_\theta(\cdot \mid s_i)$. We also include the expert action $a_i$ in our analysis.

For the expert action $a_i$, executing it leads to the next state $s_{i+1}$. For each alternative action $a_i^j \in \mathcal{A}_i$, executing it in the environment leads to a next state $s_i^j$ sampled from the transition function $T(s_i, a_i^j)$. These next states capture the immediate consequences of taking action $a_i^j$ at state $s_i$, reflecting changes in the environment such as updated DOM structures, new tool outputs, error messages, or task progression. We collect these interactions into a rollout dataset:

$$\mathcal{D}_{\text{rollout}} = \{(s_i, a_i^j, s_i^j) \mid i \in [N], j \in [K]\}, \tag{2}$$

where each triple represents a state, an alternative action taken at that state, and the resulting next state. All actions $a_i^j$ differ from the expert action $a_i$, allowing the agent to experience diverse state transitions from its own proposed actions. This rollout dataset $\mathcal{D}_{\text{rollout}}$ provides rich supervision signals without requiring explicit rewards. The next states $\{s_i^j \mid j \in [K]\}$ encode implicit feedback about action quality through environment responses, enabling the agent to learn from the consequences of both expert and non-expert behaviors.

Building on the notation from §3, we leverage the expert dataset $\mathcal{D}_{\text{expert}} = \{(s_i, a_i)\}_{i=1}^N$ and the rollout dataset $\mathcal{D}_{\text{rollout}} = \{(s_i, a_i^j, s_i^j) \mid i \in [N], j \in [K]\}$ to develop two different training approaches under the same early experience principle. The key insight is that the next states $s_i^j$ resulting from non-expert actions provide valuable supervision signals without explicit rewards. We now describe how this dataset is leveraged by our two early experience methods.

## 4.2 IMPLICIT WORLD MODELING

We formulate world modeling as an auxiliary prediction task that helps the agent internalize environment dynamics from its own early experience. In our setting, states are represented entirely in natural language, allowing us to model next-state prediction as a standard next-token prediction objective. Inspired by prior work on training LLMs as world models (Gu et al., 2025), we use next states from the rollout set $\mathcal{D}_{\text{rollout}}$ as direct training signals for the language agent's policy $\pi_\theta$. For example, when booking flights on the web, the model may predict the page state after entering an invalid date, learning from the textual error message as a natural-language representation of the next state. This design removes the need for a separate module and fits naturally the LLM fine-tuning paradigm.

For each rollout triple $(s_i, a_i^j, s_i^j) \in \mathcal{D}_{\text{rollout}}$, we construct a prediction task where the model takes the state-action pair $(s_i, a_i^j)$ as input and learns to predict the resulting next state $s_i^j$. We define the training objective as a next-token prediction loss:

$$\mathcal{L}_{\text{IWM}} = -\sum_{(s_i, a_i^j, s_i^j) \in \mathcal{D}_{\text{rollout}}} \log p_\theta(s_i^j \mid s_i, a_i^j), \tag{3}$$

where $p_\theta$ denotes the language model's output distribution. Note that we use the same model parameters $\theta$ for both state prediction (during world modeling) and action prediction (during policy execution), allowing the policy to internalize environment dynamics directly.

This training objective encourages the model to capture regularities in environment behavior, including common transitions, side effects, and invalid action outcomes. Unlike inference-time world models used for planning, our *implicit* formulation integrates predictive signals directly into policy learning, serving as a lightweight warm-up before supervised learning or downstream optimization. It exposes the agent to diverse, non-expert behaviors, improving robustness to distribution shifts and reducing dependence on brittle expert trajectories. In practice, the rollout data are often an order of magnitude larger than $\mathcal{D}_{\text{expert}}$. We adopt a two-stage pipeline: first train with $\mathcal{L}_{\text{IWM}}$ to internalize coarse dynamics, then fine-tune on $\mathcal{D}_{\text{expert}}$ (*i.e.*, $\mathcal{L}_{\text{IL}}$).

## 4.3 SELF-REFLECTION

We formulate self-reflection as a mechanism for agents to learn from their own exploratory outcomes. Rather than relying solely on expert state–action pairs, the agent compares the expert action at each state with alternatives sampled from its policy, using the resulting next states to generate natural language explanations of why the expert choice is better. These explanations provide richer, transferable supervision than expert actions alone, leveraging the LLM's strength in processing language to internalize decision principles that generalize across tasks.

Specifically, for each expert state $s_i$, we first execute the expert action $a_i$ to obtain the expert next state $s_{i+1}$. For each alternative action $a_i^j$ (where $j \in \{1, ..., K\}$), we obtain the corresponding next state

$s_i^j$. We then prompt a language model to generate a chain-of-thought $c_i^j$ explaining why the expert action $a_i$ is preferable to the alternative $a_i^j$ based on the differences between their resulting states $s_{i+1}$ and $s_i^j$. This prompt is designed to elicit natural language reasoning that highlights potential limitations or inefficiencies in $a_i^j$, grounded in the actual state transitions observed.

The resulting triplets $(s_i, a_i^j, c_i^j)$ are collected into a dataset $\mathcal{D}_{\text{refl}}$. We then train the agent to jointly predict the chain-of-thought and the expert action conditioned on the state $s_i$, using a next-token prediction loss over the concatenated target sequence $c_i^j \circ a_i$:

$$\mathcal{L}_{\text{SR}} = - \sum_{(s_i, a_i^j, c_i^j) \in \mathcal{D}_{\text{refl}}} \log p_\theta(c_i^j, a_i \mid s_i), \tag{4}$$

where $p_\theta$ denotes the language model's output distribution, aligned with the agent's policy $\pi_\theta$.

In practice, we mix the self-reflection data $\mathcal{D}_{\text{refl}}$ with the expert dataset $\mathcal{D}_{\text{expert}}$ and train the model using a standard next-token prediction loss. Chain-of-thought reasoning is generated only for the self-reflection training data, and we retain the original chain-of-thought reasoning in $\mathcal{D}_{\text{expert}}$ whenever provided by the expert trajectories, for all models trained with $\mathcal{D}_{\text{expert}}$.

Learning from both sources encourages the model to move beyond rote imitation and develop more generalizable decision criteria. For example, in `WebShop`, when the expert action is "click on the $15 blue shirt," an alternative might be "click on the $30 red shirt." The generated reflection could be: "While the red shirt matches the color preference, it exceeds the $20 budget constraint specified in the query. The blue shirt satisfies both the style requirement and budget limit." This teaches the model to prioritize constraints, a lesson that generalizes beyond this specific item. We show the prompt used across environments in § E.1

Both implicit world modeling and self-reflection follow the same principle of turning the agent's own actions and resulting future states into scalable supervision, enabling stronger language agents.

## 5 EXPERIMENTS

We evaluate the early experience paradigm via the proposed two methods under this paradigm across a diverse suite of language-agent environments, testing its effectiveness (§5.2), out-of-domain generalization (§5.3), and compatibility with post-hoc reinforcement learning (§5.4).

### 5.1 EXPERIMENT SETUP

We train and evaluate instruction-tuned Llama models ( Llama-3.1-8B, Llama-3.2-3B) on eight diverse language-agent benchmarks spanning embodied, science, planning, QA, tool-use, and web navigation tasks (Shridhar et al., 2021; Wang et al., 2022; Yao et al., 2022; Zhou et al., 2024; Xie et al., 2024a; Patil et al., 2025; Yao et al., 2025; Jin et al., 2025). All environments, data sources, model details, and the full training and evaluation protocols are provided in Appendix E.

### 5.2 EFFECTIVENESS

We evaluate across eight environments that span multi-turn tool use, web navigation, and more (Table 1). All models are trained with the same prompt format and decoding strategy and, per environment, our methods use exactly the same step budget as imitation learning.

**Overall Gains.** Early experience improves over imitation learning in nearly all settings and with both model sizes. Implicit World Modeling (IWM) yields steady gains in structured simulators and transactional sites (`ALFWorld`/`ScienceWorld` +2.3 to +5.5; `WebShop` +11.3 to +18.4). Self-Reflection (SR) delivers the largest jumps when tasks require multi-step reasoning and constraint satisfaction (`TravelPlanner` +12.8 to +15.0; `ScienceWorld` +13.3; `BFCLv3` +8.0 on the 3B model). Even on the most challenging settings, the gains are consistent though smaller in absolute terms (`WebArena` +1.2 to +3.6; `SearchQA` +0.6 to +3.3).

**Action-Space Perspective.** Across our eight environments, the action spaces fall into three regimes. *Closed and finite action sets* (*e.g.*, `ALFWorld` for embodied navigation, `ScienceWorld` for scientific procedures, and `TravelPlanner` for itinerary planning) present a small, fixed list of admissible actions from the start. Here, IWM helps the policy internalize transition regularities, while SR adds targeted corrections for long-horizon plans (*e.g.*, large SR gains on `TravelPlanner`).

Table 1: Evaluation results on eight benchmarks. All values are task-success rates (%) unless otherwise noted. Improvements over imitation learning are shown in green. Prompt indicates the performance of the instruction-tuned model. IWM and SR denote Implicit World Modeling and Self-Reflection, respectively. See Appendix F for complete results for each benchmark.

| Benchmark | Model | Prompt | Imitation Learning | Ours-IWM | Ours-SR |
|---|---|---|---|---|---|
| *Embodied and Scientific Simulation, and Travel Planning* | | | | | |
| ALFWorld | -3.2-3B | 8.6 | 78.1 | 83.6 (+5.5) | **85.9 (+7.8)** |
| | -3.1-8B | 25.0 | 80.5 | **85.9 (+5.4)** | 85.2 (+4.7) |
| ScienceWorld | -3.2-3B | 2.3 | 51.6 | 55.5 (+3.9) | **56.2 (+4.6)** |
| | -3.1-8B | 3.1 | 54.7 | 57.0 (+2.3) | **68.0 (+13.3)** |
| TravelPlanner | -3.2-3B | 0.0 | 19.4 | 28.3 (+8.9) | **32.2 (+12.8)** |
| | -3.1-8B | 0.0 | 17.2 | 25.0 (+7.8) | **32.2 (+15.0)** |
| *Multi-Turn Tool-Use* | | | | | |
| BFCLv3 | -3.2-3B | 1.3 | 21.3 | 25.3 (+4.0) | **29.3 (+8.0)** |
| | -3.1-8B | 6.7 | 16.0 | **20.0 (+4.0)** | **20.0 (+4.0)** |
| Tau-Bench | -3.2-3B | 5.2 | 24.3 | 26.1 (+1.8) | **28.7 (+4.4)** |
| | -3.1-8B | 6.0 | 35.9 | 40.8 (+4.9) | **41.7 (+5.8)** |
| SearchQA (F1) | -3.2-3B | 13.3 | 38.0 | **39.0 (+1.0)** | 38.6 (+0.6) |
| | -3.1-8B | 21.0 | 41.0 | **44.3 (+3.3)** | 41.8 (+0.8) |
| *Web Navigation* | | | | | |
| WebShop | -3.2-3B | 0.0 | 41.8 | **60.2 (+18.4)** | 52.7 (+10.9) |
| | -3.1-8B | 0.0 | 47.3 | 58.6 (+11.3) | **58.2 (+10.9)** |
| WebArena | -3.2-3B | 1.2 | 6.1 | **8.5 (+2.4)** | 7.3 (+1.2) |
| | -3.1-8B | 0.6 | 4.9 | **8.5 (+3.6)** | 7.9 (+3.0) |

*Structured but large action sets* (*e.g.*, BFCLv3 for terminal tasks and Tau-Bench for multi-domain APIs) require selecting from many typed tools with arguments and sequencing them correctly. In this setting, early experience reduces tool misuse and improves ordering; SR often helps more when policy errors are logical. *Open action sets* (*e.g.*, SearchQA with free-form search queries, WebArena with fine-grained web element interactions) allow a vast number of possible actions, often combinatorial in nature. These are the hardest regimes; nevertheless, early experience still yields reliable gains by turning exploratory rollouts into dense training signals without requiring rewards.

**Observation-Space Perspective.** Our benchmarks span a wide range of observation complexities. At the low end, ALFWorld provides short, clean textual descriptions of the scene, while ScienceWorld produces procedural readouts of ongoing experiments. Mid-range settings like BFCLv3 and Tau-Bench return structured API schemas and tool outputs that must be parsed and sequenced correctly. At the high end, WebArena presents noisy, fine-grained web states as accessibility trees, requiring reasoning over hundreds of DOM-like elements. We provide examples of each environment in Appendix F. In settings where state transitions are consistent and predictable (*e.g.*, WebShop), IWM excels by helping the agent internalize environment dynamics and improve next-state predictions. When failures stem primarily from reasoning errors or the need to repair long-horizon plans (*e.g.*, TravelPlanner, ScienceWorld), SR delivers larger gains by explicitly comparing actions to expert trajectories. Overall, regardless of how simple or complex the environment's observations are, early experience methods consistently turn the agent's own actions and resulting states into effective supervision signals that improve policy learning without rewards.

**Takeaway.** Early experience reliably converts an agent's own actions and resulting states into scalable supervision beyond expert demonstrations. Both methods under this paradigm strengthen policies across environments that differ substantially in both action spaces and observation complexity. These effects hold across two model sizes and three environment families, demonstrating strong generalizable feasibility of our early experience paradigm.

## 5.3 Out-Of-Domain Generalization

To evaluate the robustness of trained policies beyond in-domain performance, we explore early experience in environments with out-of-domain (OOD) splits, using the same checkpoints evaluated

Table 2: Out-of-domain evaluation results (%). Improvements over imitation learning are shown in green. Prompt means the instruct model's performance. IWM and SR refers to Implicit World Modeling and Self-Reflection, respectively.

| | AlfWorld | | BFCLv3 | | SearchQA (F1) | |
|---|---|---|---|---|---|---|
| | -3.2-3B | -3.1-8B | -3.2-3B | -3.1-8B | -3.2-3B | -3.1-8B |
| Prompt | 5.5 | 18.8 | 1.3 | 6.2 | 31.9 | 40.0 |
| Imitation Learning | 74.2 | 63.3 | 5.3 | 6.7 | 40.5 | 47.4 |
| Ours-IWM | **77.3** (+3.1) | **78.1** (+14.8) | 8.9 (+3.6) | 7.6 (+0.9) | **45.4** (+4.9) | 49.6 (+2.2) |
| Ours-SR | **77.3** (+3.1) | 72.7 (+9.4) | **13.8** (+8.5) | **8.0** (+1.3) | 44.0 (+3.5) | **50.7** (+3.3) |

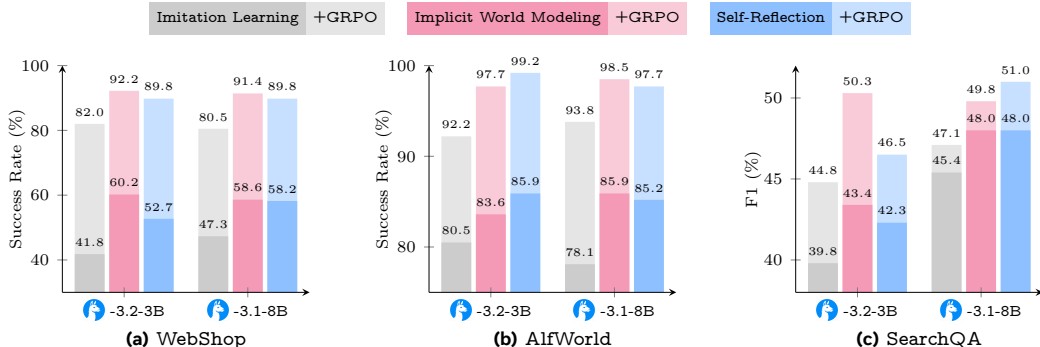

Figure 3: Reinforcement learning (GRPO) starting from checkpoints trained with different methods on three infra-ready environments. Bars show performance before (deeper shade) and after RL (lighter shade) for three methods. Checkpoints from IWM and SR consistently lead to higher post-RL ceilings than imitation-only starts, with advantages often maintained or amplified after RL.

in Section 5.2. To setup, for `ALFWorld` and `SearchQA` we follow the OOD splits defined in their original work. For `BFCLv3` the in-domain setting is multi-turn *base*; OOD settings are averaged over multi-turn *missing function*, *missing argument*, and *long context*.

The results of our trained models are shown in Table 2, from which we can make the following observations. OOD scores drop relative to in-domain across all tasks, yet early experience consistently recovers a substantial portion of the gap. In several cases the relative gains are larger than in-domain (*e.g.*, `SearchQA`), indicating that converting one's own rollouts into supervision prepares the policy for states not covered by demonstrations. The method-wise pattern mirrors in-domain trends: IWM helps most where dynamics are stable (*e.g.*, `ALFWorld`); SR is strongest when distribution shifts alter tool availability or arguments (*e.g.*, `BFCLv3`); both IWM and SR help under retrieval shifts (*e.g.*, `SearchQA`), for both model sizes.

**Takeaway.** Early experience improves robustness under diverse OOD regimes: IWM excels when dynamics are stable, SR when shifts affect tool availability, arguments, or retrieval distributions. In several benchmarks (*e.g.*, `ALFWorld`, `SearchQA`), OOD gains meet or exceed in-domain gains, reinforcing that an agent's own experience provides supervision that generalizes.

## 5.4 REINFORCEMENT LEARNING FOLLOWING EARLY EXPERIENCE

To evaluate the impact of early experience once environments provide verifiable rewards (the defining condition of the *era of experience*), we append a reinforcement learning stage to models trained in Section 5.2. We focus on three reward-available benchmarks: `WebShop`, `ALFWorld`, and `SearchQA`, and adopt the widely used GRPO algorithm (Shao et al., 2024) with identical hyperparameters and training steps as established recipes (Feng et al., 2025; Jin et al., 2025). The only factor that changes across runs is the initialization: Imitation Learning (IL), Implicit World Modeling (IWM), or Self-Reflection (SR).

Results in Figure 3 show a clear pattern: starting from early experience consistently yields higher post-RL ceilings. In some cases, the performance gap grows during RL training (*e.g.*, `ALFWorld`); in others, it narrows but never reverses. Even when reward optimization is applied for the same number

of steps, IL starts rarely match the final performance of early-experience starts. For completeness, we also run GRPO directly from the raw pretrained model without any supervised stage. This performs worst across all tasks and shows unstable training dynamics, highlighting the necessity of a strong initialization. The full results with detailed metrics can be found in Appendix E.

**Takeaway.** Early experience acts as a *mid-training bridge* between the era of human data and the era of experience. It produces policies that already perform strongly without rewards and that amplify the benefits of subsequent RL. Under identical RL recipes, early-experience starts achieve higher final performance. These results suggest that once RL setups become available in new environments, early experience can immediately unlock further gains without retraining from scratch.

## 6 DISCUSSION

We make several discussions in terms of the data amount (§ 6.1) and branching fact (§ 6.2) in this section, as well as data synthesis baselines (§ D.1) and model scaling (§ D.2) in the Appendix D.

### 6.1 IMPACT OF AMOUNT OF HUMAN DATA

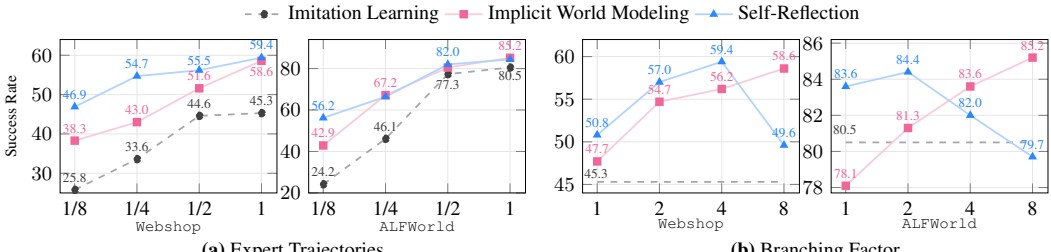

**(a)** Expert Trajectories        **(b)** Branching Factor

Figure 4: Effect of demonstration budget and branching factor. **(a)**: success rate vs. fraction of expert trajectories; **(b)**: success rate vs. branching factor $K$ (number of alternative actions per state in $\mathcal{D}_{\text{expert}}$). Results for `WebShop` and `ALFWorld` using Llama-3.1-8B-Instruct.

To examine how performance scales with the amount of expert supervision, we vary the number of demonstrations used to seed early experience while keeping the total training budget fixed. Figure 4 (a) shows that early experience maintains a consistent lead over imitation learning at every data level. On `WebShop`, just $1/8$ of the demonstrations already surpasses imitation learning trained on the full dataset; on `ALFWorld`, the same holds with $1/2$ of the demonstrations. Both IWM and SR improve with more expert data, yet the margin over imitation learning remains large, underscoring that early experience provides additional supervision signals beyond what demonstrations alone can supply.

### 6.2 IMPACT OF BRANCHING FACTOR

To investigate the impact of branching factor for our methods, we also ablate the branching factor $K$, the number of alternative actions rolled out per expert state when generating early experience. Figure 4 (b) shows that IWM improves steadily as $K$ increases, consistent with learning richer transition regularities. SR improves at small to moderate $K$ and can be non-monotonic at very large $K$: comparing many alternatives occasionally includes other success-leading actions, reducing contrast with the expert, and current models have limited capacity to reason over many alternatives and outcomes in a single context. Overall, both variants improve most of the time, with IWM favoring larger $K$ and SR working best with a modest $K$ (*e.g.*, 2–4).

## 7 CONCLUSION

We advocate and present *early experience* as a scalable, reward-free paradigm that advances language agents before reinforcement learning environments are fully ready. By converting an agent's own actions and resulting states into supervision, without external reward signals, we achieve consistent gains across eight diverse environments, spanning embodied navigation, scientific experimentation, long-horizon planning, multi-turn tool use, and web navigation. The proposed two methods under this paradigm, implicit world modeling and self-reflection, improve both in-domain effectiveness and out-of-domain robustness, and retain their advantage when used to warm-start reinforcement learning, positioning early experience as a practical and general foundation for building more capable language agents in the upcoming *era of experience*.

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

APPENDICES

OVERVIEW

Our supplementary includes the following sections:

- **Section A: Statement on Large Language Model Usage.** We disclose how large language models were used in this work.
- **Section B: Limitation and Future Work.** We discuss the Limitations of early experience and potential future work.
- **Section C: More Related Work.** We discuss more related works in terms of our proposed method such as self-reflection and world modeling.
- **Section D: More Discussions on Our Method.** We make discussions about other potential data synthesis baselines and model scaling impact.
- **Section E: Implementation Details.** We provide implementation details on the prompt and experiment setup.
- **Section F: Full Results.** We detail the results of each benchmark and we provide concrete case studies for each benchmark and our method.

## A    STATEMENT ON LARGE LANGUAGE MODEL USAGE

Large language models (LLMs) were used solely to polish the text and improve linguistic clarity, and to generate auxiliary code for rendering figures and other visuals. They did not contribute to the research methodology, experimental design, data analysis, or interpretation of results.

## B    LIMITATIONS AND FUTURE WORK

While early experience improves performance across diverse environments, several limitations remain. Our current approaches, implicit world modeling and self-reflection, focus on short-horizon traces; extending them to address long-horizon credit assignment without explicit rewards remains an open challenge. In addition, we have not yet unified training across multiple environments to fully realize the potential of early experience, partly due to limitations in our current training infrastructure.

Future work will explore combining early experience with richer self-supervised objectives, leveraging cross-environment transfer, and integrating it with reward-based fine-tuning in a continual learning setting. Another direction is to investigate other instances of early experience beyond the two approaches proposed in this paper. We also hope to extend the paradigm to large-scale, real-world deployments, where interaction data is collected organically and can drive continual policy improvement.

## C    MORE RELATED WORKS

### C.1    SUPERVISION FROM EXPLORATION

Traditional exploration–exploitation strategies in RL collect trajectories that are later refined through reward feedback. Methods like experience replay (Andrychowicz et al., 2017) densify sparse rewards by retrofitting achieved outcomes as goals, but still require verifiable reward functions unavailable in many language agent environments. Our setting uses exploration differently: interaction traces become direct supervision signals, eliminating the need for rewards or manual relabelling entirely.

**World Models.** World models (Sutton, 1991; Ha & Schmidhuber, 2018; Hafner et al., 2020; 2021) are traditionally trained on observed state transitions to predict future states and rewards, allowing model-based RL to reduce sample complexity and support speculative planning. Recent work extends this idea to language agents by using LLMs as world models (Gu et al., 2025; Chae et al., 2025; Hao et al., 2023), which improves downstream performance through language-mediated simulations. Despite the different state representations of world models in different era, most of these systems still treat the world model as a *separate* simulator, echoing classical control pipelines. In contrast, we view the interaction trace itself as an auxiliary prediction task for the agent policy, similar in spirit to mid-training (Zhang et al., 2025). By training the policy to predict its own future states, the model internalizes coarse environment dynamics without a standalone simulator. This *implicit* world model

grounds the agent in its operating context, offers a lightweight warm-up for faster adaptation, and avoids the planning overhead required by explicit simulators.

**Self-Reflection.** Self-reflection (Shinn et al., 2023; Madaan et al., 2023) was initially introduced as a prompting technique that allows LLMs to revise their answers through multi-turn self-dialogues (Snell et al., 2024) or curated prompt variants (Madaan et al., 2023), without updating model parameters. Subsequent work summarizes lessons over rewarded trajectories in the prompt (*e.g.*, short-term episodic memory (Xie et al., 2025)) to guide future inference. However, later studies (Huang et al., 2024; Valmeekam et al., 2023) show that such inference-time methods often fail without access to external feedback (*e.g.*, rewards). A separate line uses LLMs to generate rationales for correct answers, treating these rationales as training targets to boostrap reasoning (Zelikman et al., 2022; Huang et al., 2023). We extend this view of reflection to the agent setting where *explicit rewards are absent*. Our approach trains agents to reflect on their own suboptimal actions and the resulting trajectories, then uses the reflected rationales as training signals to improve decision-making.

# D  MORE DISCUSSIONS

## D.1  COMPARISON TO BASELINES

Table 3: Comparison of early experience with three representative baselines. All results are based on 🤗 `Llama-3.1-8B-Instruct`.

|  | WebShop | ALFWorld |
| --- | --- | --- |
| Prompt | 0.0 | 25.0 |
| +Long CoT | 1.6 (+1.6) | 28.4 (+3.4) |
| Imitation Learning | 47.3 | 80.5 |
| +Long CoT | 0.0 (-47.3) | 25.8 (-54.7) |
| +STaR | 25.0 (-22.3) | 74.2 (-6.3) |
| Ours-IWM | 58.6 (+11.3) | 85.9 (+5.4) |
| Ours-SR | 58.2 (+10.9) | 85.2 (+4.7) |

We compare early experience to two alternatives that inject extra supervision or reasoning signals *without* executing alternative actions or observing their resulting states. This allows us to test whether our gains can be matched by simply extending reasoning at inference or by adding ungrounded rationales during training.

**(1) Long CoT** (test-time scaling). Inspired by test-time scaling (Snell et al., 2024), we aim to help instruction-tuned and imitation-only models trained on expert trajectories, where rationales are often absent, reason more extensively at inference. The prompt baseline uses the off-the-shelf instruction-tuned model with the official prompts from prior work, which typically produce short chain-of-thought (Wei et al., 2022). Our Long CoT variant forces longer reasoning before action generation by performing heavier prompt search on the training split and, when a delimiter token marking the end of reasoning exists (*e.g.*, `</think>`), truncating it to encourage continued generation. We report the best results for each model.

**(2) STaR-style data** (reasoning without alternative actions or resulting states). Following STaR (Zelikman et al., 2022), we have the model generate a rationale for the *expert* action at each state and retain only cases where the predicted action matches the expert. We then fine-tune on (state, rationale, action) tuples, as in Equation 4. Since alternative actions and their resulting states are *not* used, these rationales remain ungrounded in actual outcomes. We search over prompt variants for rationale synthesis and keep the strongest configuration. The number of optimization steps is matched to imitation learning.

Table 3 shows that both early experience methods achieve the largest gains across tasks and model sizes. For Long CoT, heavier prompt search and reasoning-length control can modestly improve the imitation-trained prompt baseline, but the gains vanish quickly in harder settings. Once fine-tuned only on expert trajectories lacking inherent rationales, models lose the ability to sustain coherent long-form reasoning, so extended chains often drift or collapse into invalid/off-policy actions despite truncation at the thought–action boundary. For STaR-style data, the match rate between generated and expert actions is low, leaving little usable training data. The retained rationales are ungrounded—having never been tested in the environment—and frequently hallucinate tools or facts, so fine-tuning on them can even degrade performance. In contrast, early experience directly converts the policy's own off-

expert rollouts into *grounded* supervision from observed next states, producing robust improvements that these alternatives fail to match.

## D.2 MODEL SCALING

We study whether the benefits of early experience persist as models scale. On `WebArena` we compare 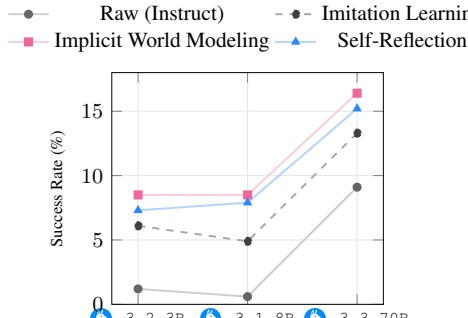 `-3.2-3B`, `-3.1-8B`, and `-3.3-70B`. Due to limited compute, fine-tuning for 70B models uses parameter-efficient LoRA (Hu et al., 2022) for all methods with the same rank and update steps; for IWM, the same adapters are continued in the second stage so that total tunable parameters and compute match imitation learning.

Figure 5 shows that early experience outperforms imitation learning at every scale, with the gap persisting even for the 70B model. Absolute performance rises with scale, and early-experience checkpoints consistently occupy the top curve, indicating that the supervision it provides comple-

Figure 5: Performance of Llama with different model sizes trained with imitation learning and methods under *early experience* on the `WebArena` benchmark.

ments model size rather than substituting for it. Even with LoRA-only updates, both IWM and SR deliver steady gains, demonstrating that the approach remains effective under constrained compute budgets.

# E  IMPLEMENTATION DETAILS

## E.1  SELF-REFLECTION PROMPT TEMPLATE

---

**Self-Reflection Prompt Template**

You will be presented with a situation where you need to choose between multiple possible actions. Your task is to analyze the situation and provide reasoning about why we decide to take the expert action.

- **Situation Description ($s_i$):** {Situation Description}

- **Expert Action ($a_i$):** {Expert Action}

- **Expected Outcome ($s_{i+1}$):** {Future State of Expert Action}

- **Alternative Actions:**

  1. Action $a_i^1$: {Alt Action 1}, resulting state $s_i^1$: {Obs 1}
  2. Action $a_i^2$: {Alt Action 2}, resulting state $s_i^2$: {Obs 2}
  3. ...

Provide a detailed self-reflection as an *internal monologue* that demonstrates your reasoning process for the current situation. Your monologue should:

1. Analyze the situation and the goal.

2. Compare the possible actions, explaining why each may be less optimal.

3. Justify why the expert action is most suitable, grounded in the expected outcome.

4. Highlight any relevant clues, constraints, or consequences from the situation.

**Guidelines:**

- Stay strictly within the provided information.

- Avoid meta-commentary about being an AI.

- Use natural, step-by-step reasoning.

- Focus on logical decision-making.

**Output:** Directly write the self-reflection monologue, no extra headings, disclaimers, or external notes.

---

## E.2  DETAILED EXPERIMENT SETUP

**Environments.** We conduct experiments on eight language-agent environments covering a wide range of domains and task formats: *1)* `ALFWorld` (Shridhar et al., 2021): embodied instruction-following tasks in a simulated household, combining textual descriptions with high-level symbolic actions. We follow the setting of Feng et al. (2025). *2)* `ScienceWorld` (Wang et al., 2022): an interactive science lab simulator rendered in natural language, where agents perform multi-step experiments using tools and materials. We implement the gym (Brockman et al., 2016) for this environment. *3)* `TravelPlanner` (Xie et al., 2024a): long-horizon travel planning tasks that require generating and refining multi-day itineraries using various tools and databases. We focus on the sole-planning mode and implement the gym for such an environment. *4)* `SearchQA` (Jin et al., 2025): multi-hop question answering in open-domain settings, where agents issue search queries and reason over retrieved snippets to answer complex questions. We follow Search-R1 (Jin et al., 2025) settings. *5)* `BFCLv3` (Patil et al., 2025): multi-turn tool-use tasks from the Berkeley Function Call Leaderboard v3, where agents interact with a Python-based API environment that simulates functional programs. We focus on the multi-turn tool use. *6)* `Tau-Bench` (Yao et al., 2025): realistic customer-service scenarios requiring agents to interact with LM-simulated users, perform multi-turn tool use via APIs, and adhere to domain-specific policy documents. We focus on the Retail subset. *7)* `WebShop` (Yao et al., 2022): goal-oriented shopping tasks in a simulated e-commerce site, where agents must navigate, filter, and select the correct product based on natural

language queries. We follow the setting of Feng et al. (2025). *8)* `WebArena` (Zhou et al., 2024; Liu et al., 2025): web navigation tasks across domains like e-commerce, forums, and content management, with embedded tools and knowledge bases. We follow Koh et al. (2024) to evaluate web arena results with accessibility tree as observation space.

**Models and Expert Trajectories.** We evaluate early experience using two instruction-tuned models from the Llama family: `Llama-3.1-8B` and `Llama-3.2-3B`, each trained using a fixed number of expert demonstrations with or without early experience augmentation. These demonstrations are drawn from different sources across environments: *1)* directly provided optimal trajectories (`ALFWorld` and `ScienceWorld`); *2)* successful but potentially suboptimal human-/model-collected trajectories (`WebShop` and `WebArena`); *3)* LLM-assisted data synthesis workflows on the training split, where demonstrations are absent (`SearchQA`, `TravelPlanner`, `BFCLv3`, and `Tau-Bench`);

**Training and Evaluation.** We use consistent prompt formatting and decoding strategies across all settings. Because environments differ in data size and horizon, we first explore the number of optimization steps for the Imitation Learning baseline in each environment and select the checkpoint by lowest training loss as well as the performance on the validation set. We then fix this step budget and use it unchanged for our methods to ensure a fair comparison. For Implicit World Modeling, we begin with one epoch of the WM objective and then continue supervised updates so that the total updates equal the imitation budget without extra steps. For Self-Reflection, we train for the same number of epochs as imitation. All experiments use at most 8 H100 GPUs for training and evaluation. In terms of evaluation, we report each benchmark's main native metric and follow its official validators. For full evaluation results, please refer to Appendix E.

# F    Full Results

In this section, we provide full results for each environment. For each one, we present tables containing all available metrics. Also, we show concrete training examples for and synthesized (*e.g.*, for self-reflection) by `Llama-3.1-8B`.

## F.1    ALFWORLD

We follow the default split of `ALFWorld` (Shridhar et al., 2021) with the `TextWorld` (Côté et al., 2019) setup under the Verl-Agent (Feng et al., 2025) framework. From the expert trajectories in `ALFWorld`, we extract 21,031 state–action pairs to form $\mathcal{D}_{\text{expert}}$ These expert trajectories are optimal given the completeness of task solvability in the dataset.

For implicit world modeling, we augment $\mathcal{D}_{\text{expert}}$ with $\mathcal{D}_{\text{rollout}}$. At each state, we sample 8 non-expert actions uniformly without replacement from the admissible action list (excluding the expert action) and include the expert action, yielding $21{,}031 \times 9 = 189{,}279$ triplets for implicit world modeling.

For self-reflection, we construct data by prompting the model to explain its own decisions. For each state, we use the same policy model with temperature 1.0 to propose up to 3 alternative actions. We canonicalize proposed actions and keep only unique ones. If a proposed action is not in the admissible action space for that state, we discard it and instead sample uniformly at random from the remaining unselected admissible actions. The final prompt asks the model to justify why the expert action is preferable to the sampled alternatives given the current state and available tools.

During training, we use a batch size of 16 and a learning rate of $1\mathrm{e}{-5}$, and train with `LlamaFactory` (Zheng et al., 2024b) for 2 epochs. For RL training, we adopt the default hyperparameters in Verl-Agent and evaluate on the same split reported in their paper. For the evaluation, we set the maximum prompt length to be 4096, the maximum response length to be 1024, and the temperature to be 0.4. Below is a data example illustrating the inputs for different methods given the same task and state (we list three examples for implicit world modeling). Full results are in Table 4.

| **Training Example — `ALFWorld`** | |
|---|---|
| **Task:** | put two book in bed. |
| **Current State:** | You have taken the action 1: 'go to desk 1' You are now at step 2 and your current observation is: You arrive at desk 1. On the desk 1, you see a book 3, a book 2, a book 1, a bowl 1, a cd 1, a cellphone 2, a creditcard 2, a mug 1, a statue 2, and a statue 1. |
| **Expert Action:** | take book 3 from the desk 1 |
| **Implicit World Modeling:** | • examine desk 1 → "Nothing happens."
• examine drawer 1 → "Nothing happens."
• inventory → "You are not carrying anything." |
| **Self-Reflection:** | Let me think through the best course of action. The goal is to put two books in the bed. I have just arrived at *desk 1* and can see three books. Examining the desk or drawers does not surface new items or constraints, so those actions are unhelpful at this point. The most direct way to make progress is to start collecting the required objects. **Taking book 3 from desk 1** immediately moves me toward the objective without extra navigation or redundant examinations. After securing the first book, I can take a second one from the same location, then navigate to the bed and place both books there. This sequence minimizes detours and aligns tightly with the task. |

Table 4: Performance on ALFWorld. Results of closed-source models are adopted from Feng et al. (2025).

| Type | Method | ALFWorld | | | | | | |
|---|---|---|---|---|---|---|---|---|
| | | Pick | Look | Clean | Heat | Cool | Pick2 | All |
| *Base: Closed-Source Model* | | | | | | | | |
| Prompting | GPT-4o | 75.3 | 60.8 | 31.2 | 56.7 | 21.6 | 49.8 | 48.0 |
| Prompting | Gemini-2.5-Pro | 92.8 | 63.3 | 62.1 | 69.0 | 26.6 | 58.7 | 60.3 |
| *Base:* 🦙 `Llama-3.2-3B-Instruct` | | | | | | | | |
| Prompting | | 25.0 | 0.0 | 3.7 | 0.0 | 0.0 | 7.7 | 8.6 |
| +RL (GRPO) | | 93.3 | 60.0 | 94.7 | 82.6 | 78.3 | 52.2 | 78.9 |
| Imitation Learning | Behavior Cloning | 78.1 | 71.4 | 85.2 | 82.4 | 89.5 | 61.5 | 78.1 |
| + RL (GRPO) | | 97.4 | 77.8 | 87.5 | 100.0 | 88.9 | 91.3 | 92.2 |
| Early Experience | Implicit World Modeling | 87.5 | 85.7 | 85.2 | 88.2 | 89.5 | 69.2 | 83.6 |
| + RL (GRPO) | | 100.0 | 100.0 | 100.0 | 93.3 | 95.0 | 95.5 | 97.7 |
| Early Experience | Self-Reflection | 90.6 | 85.7 | 81.5 | 88.2 | 89.5 | 80.8 | 85.9 |
| + RL (GRPO) | | 100.0 | 100.0 | 95.0 | 100.0 | 100.0 | 100.0 | 99.2 |
| *Base:* 🦙 `Llama-3.1-8B-Instruct` | | | | | | | | |
| Prompting | | 37.5 | 28.6 | 25.9 | 11.8 | 21.1 | 19.2 | 25.0 |
| +RL (GRPO) | | 97.3 | 80.0 | 90.0 | 85.7 | 88.2 | 56.0 | 83.6 |
| Imitation Learning | Behavior Cloning | 90.6 | 85.7 | 85.2 | 82.4 | 89.5 | 53.8 | 80.5 |
| + RL (GRPO) | | 95.0 | 88.9 | 100.0 | 100.0 | 100.0 | 95.5 | 93.8 |
| Early Experience | Implicit World Modeling | 87.5 | 57.1 | 88.9 | 82.4 | 94.7 | 84.6 | 85.9 |
| + RL (GRPO) | | 100.0 | 100.0 | 100.0 | 92.9 | 100.0 | 92.0 | 97.7 |
| Early Experience | Self-Reflection | 87.5 | 71.4 | 85.2 | 82.4 | 94.7 | 80.8 | 85.2 |
| + RL (GRPO) | | 100.0 | 100.0 | 95.0 | 100.0 | 100.0 | 95.5 | 98.5 |

## F.2 WEBSHOP

From the official human demonstrations released by `WebShop` (Yao et al., 2022), we extract 1,571 human trajectories and convert them into the `Verl-Agent` (Feng et al., 2025) format, resulting in 15,464 state–action pairs that constitute $\mathcal{D}_{\text{expert}}$ for imitation learning.

For implicit world modeling, the data has two components. The first is directly derived from $\mathcal{D}_{\text{expert}}$ by reformatting each step into the world-modeling format, where the input contains the historical context and the action taken at the current step, and the target is an offline textual summary of the next state after executing that action (avg. length 345 characters). The second component is obtained by augmenting each expert state with non-expert actions: we let the same policy propose actions at temperatures $\{0.5, 0.8, 0.9\}$ and additionally sample up to five admissible actions uniformly at random per state. We then convert the augmented samples into the same world-modeling format as the first component: for each non-expert action, we execute it in the WebShop environment to obtain the subsequent observation and derive an offline textual summary of the next state. All candidates are canonicalized and deduplicated. Merging these with the expert action yields 122,954 triplets for implicit world modeling.

For self-reflection, we construct prompts that include the expert action together with 3 alternative actions and ask the model to justify why the expert action is preferable given the current state and the admissible actions. Because some actions in the raw expert trajectories are suboptimal, we apply a simple quality filter that retains only actions from trajectories whose tasks can be completed within fewer than 15 steps, resulting in 6,235 reflection examples. For each such state, the alternatives are drawn using the same policy as in world modeling by mixing model-proposed actions (at the temperatures above) with uniformly sampled admissible actions; after canonicalization and deduplication, we keep 3 distinct alternatives. We intentionally retain a diverse set of alternatives, including admissible but unhelpful actions, empty responses, and occasional invalid actions, to help the model learn sharper decision boundaries.

During training, we use a batch size of 4 and a learning rate of $1e{-}5$, and train with `LlamaFactory` (Zheng et al., 2024b). For RL training, we adopt the default hyperparameters in Verl-Agent and evaluate on the same split reported in their paper. Below is a data example illustrating the inputs for different methods given the same task and state (we list 3 examples for implicit world modeling). Full results are in Table 5.

| Training Example — `WebShop` | |
|---|---|
| **Task:** | blue wireless bluetooth headphones, and price lower than 130.00 dollars. |
| **Current State:** | you have already taken the action 1: 'search[bluetooth blue headphones]' and action 2: 'click[b082hmmbh1]'.
You are now at step 3 and your current observation is: 'Back to Search', '¡ Prev', 'color', 'black — red', 'pink', 'purple', 'black — orange', 'blue with ears', 'n-red', 'non-ears blue', 'non-ears pink', 'pink — teal', 'purple — teal', 'Bluetooth Headphones for Kids, PowerLocus Wireless Bluetooth Headphones, Kid Headphones Over-Ear with LED, Foldable Headset with Mic,Volume Limited, Wireless/Wired Headphone for Phones,Tablets,Laptop', 'Price: $24.99', 'Rating: N.A.', 'Description', 'Features', 'Reviews', 'Buy Now'. |
| **Expert Action:** | click[non-ears blue] |
| **Implicit World Modeling:** | • click[non-ears blue] → "After clicking on a color option, this page is a product-details page. The page displays various product attributes such as color options, size options, and package quantity selections, along with sections for product description, features, reviews, and a "Buy Now" button. Navigation controls include a "Back to Search" link and a "¡ Prev" button to return to the previous page."
• click[¡ prev] → "After clicking the "¡ Prev" button, this page is a search-results page displaying multiple product listings for Bluetooth headphones and related accessories. The page includes navigation controls such as "Back to Search" and "Next ¿," along with a list of product items showing titles, prices, and product IDs. Each listing is accompanied by a clickable option to view more details."
• click[buy now] → "After clicking "Buy Now" on the product details page, this page is a checkout confirmation or thank-you page. It contains a message thanking the user for their purchase, a section for entering or pasting a code, and displays the purchased item's ASIN along with selected options such as color. The page also includes a reward score and details, though the specifics of the reward are not elaborated. Navigation options are limited to a search function at this stage." |
| **Self-Reflection:** | I will now proceed with step-by-step reasoning. The task constraints are: select blue wireless Bluetooth headphones priced lower than $130.00. The current situation is that you are on a product details page for a specific item, which includes a color selection menu with multiple options, a product title, price, and rating information, along with tabs for viewing the product description, features, and reviews, as well as a "Buy Now" button. The navigation options include returning to the search or going back to the previous page. Click[non-ears blue] wins because it directly addresses the color requirement and allows for further evaluation of the product details. Other actions fail because they either do not address the color requirement or may lead to irrelevant results, such as selecting a different color, returning to the search results page without making a color selection, or selecting a blue color option with ears which may not be the desired style. **Therefore, click[non-ears blue] is optimal.** |

Table 5: Performance on WebShop.

| Type | Method | WebShop | |
|---|---|---|---|
| | | score | succ. |
| *Base:* Llama-3.2-3B-Instruct | | | |
| Prompting | | 1.3 | 0 |
| +RL (GRPO) | | 17.2 | 3.9 |
| Imitation Learning | Behavior Cloning | 55.1 | 41.8 |
| + RL (GRPO) | | 89.4 | 82.0 |
| Early Experience | Implicit World Modeling | 71.9 | 60.2 |
| + RL (GRPO) | | 97.9 | 92.2 |
| Early Experience | Self-Reflection | 67.2 | 52.7 |
| + RL (GRPO) | | 93.8 | 89.8 |
| *Base:* Llama-3.1-8B-Instruct | | | |
| Prompting | | 0 | 0 |
| +RL (GRPO) | | 2.1 | 0.8 |
| Imitation Learning | Behavior Cloning | 66.8 | 47.3 |
| + RL (GRPO) | | 90.9 | 80.5 |
| Early Experience | Implicit World Modeling | 72.7 | 58.6 |
| + RL (GRPO) | | 96.0 | 91.4 |
| Early Experience | Self-Reflection | 72.5 | 58.2 |
| + RL (GRPO) | | 94.1 | 89.8 |

### F.3 BFCLv3

We follow the default multi-turn function call split of the `BFCLv3` (Patil et al., 2025) benchmark, which categorizes tasks into *Base*, *Long-Context*, *Miss Function*, and *Miss Parameters*. *Base* contains foundational yet diverse multi-turn interactions, where all necessary information, including the user request, execution results from previous turns, and exploratory function outputs, is available to complete the task without ambiguity. *Long-Context* evaluates the model's ability to maintain accuracy in lengthy, information-dense settings by introducing large amounts of extraneous data (*e.g.*, hundreds of files or thousands of records), thereby testing its capacity to extract essential details under cognitive load. *Miss Function* assesses whether the model can identify when no available function can fulfill the user request; once this limitation is recognized, the missing functions are provided in later turns, requiring the model to adapt to newly available capabilities. *Miss Parameters* examines whether the model can detect when essential parameters are absent from the user request and cannot be inferred from the system state, prompting it to request clarification rather than making unwarranted assumptions.

As the default BFCLv3 benchmark does not provide training split, for constructing the training set, we exclusively use samples from the *Base* category. We randomly select 75% of them (125 trajectories) as the expert trajectories $\mathcal{D}_{\text{expert}}$ for imitation learning. Each trajectory consists of multiple steps and interactions, which we further split into individual steps to improve training efficacy.

For implicit world modeling, the data has two components. The first is directly derived from $\mathcal{D}_{\text{expert}}$ by reformatting each trajectory into the world-modeling format, where given the historical context and the action at the previous step, the model predicts the next state. This yields 1,264 training examples. The second is generated through augmentation: for each state in the expert trajectory, we let target model sample ten alternative actions in addition to the expert action, producing 11,904 samples, following the same process as in ALFWorld.

For self-reflection, we construct training data by prompting the model to explain its decisions, emphasizing why the expert action is preferable to other available actions in the current state, including the set of tools defined earlier. After filtering a small number of low-quality samples where the concluded action did not match the expert action, we obtain 1,200 training examples.

We train with a batch size of 16, a learning rate of $1e-5$, using `LlamaFactory` (Zheng et al., 2024b). For inference, we adopt the `VLLM` infrastructure for efficiency. A data example showing inputs for different methods under the same task and state is given below. Full results are in Table 6.

| Training Example — `BFCLv3` | |
|---|---|
| **Task:** | Go to the 'workspace' directory and move one of the 'log.txt' files into a new directory 'archive'. |
| **Current State:** | Located at the 'workspace' directory with access to the following tools: 'cat', 'cd', 'cp', 'diff', 'du', 'echo', 'find', 'grep', 'ls', 'mkdir', 'mv', 'pwd', 'rm', 'rmdir', 'sort', 'tail', 'touch', 'wc'. |
| **Expert Action:** | Move the 'log.txt' file to the 'archive' directory. |
| **Implicit World Modeling:** | • 'ls' → "Find that the 'archive' folder existed. "
• 'mv' → " 'mv' tool can move the targeted file to the targeted folder "
• 'cat', 'cd', 'cp', 'diff', 'du', 'echo', 'find', 'grep', 'mkdir', 'pwd', 'rm', 'rmdir', 'sort', 'tail', 'touch', 'wc'. → "Cannot help fulfill the user's task." |
| **Self-Reflection:** | Given the user's request to move the 'log.txt' file into a new directory called 'archive', the most appropriate action is to use the 'mv' command. This action directly achieves the goal without any unnecessary steps. The 'rm' command would remove the file instead of moving it, which is not what the user wants. Creating a directory using 'mkdir' is redundant since 'archive' already exists. Sorting the file or listing its contents won't help in archiving it. Therefore, **moving the file to the 'archive' directory is the best course of action**. |

Table 6: Performance on BFCLv3 multi-turn scenarios.

| Type | Method | I.D. Base | OOD Long Cont. | Miss Func. | Miss Para. | Avg. |
|---|---|---|---|---|---|---|
| *Base:* Llama-3.2-3B-Instruct | | | | | | |
| Prompting | | 1.3 | 1.3 | 1.3 | 1.3 | 1.3 |
| Imitation Learning | Behavior Cloning | 21.3 | 9.3 | 0.0 | 6.7 | 9.3 |
| Early Experience | Implicit World Modeling | 25.3 | 13.3 | 1.3 | 12.0 | 13.0 |
| Early Experience | Self-Reflection | 29.3 | 21.3 | 5.3 | 14.7 | 17.7 |
| *Base:* Llama-3.1-8B-Instruct | | | | | | |
| Prompting | | 6.7 | 6.7 | 8.0 | 4.0 | 6.8 |
| Imitation Learning | Behavior Cloning | 16.0 | 8.0 | 1.3 | 10.7 | 9.0 |
| Early Experience | Implicit World Modeling | 20.0 | 8.0 | 4.0 | 10.7 | 10.7 |
| Early Experience | Self-Reflection | 20.0 | 17.3 | 0.0 | 6.66 | 11.0 |

### F.4 TAU-BENCH

We conduct experiment using the retail task from `Tau-Bench`. In `Tau-Bench`, the retail task is divided into a training set and an evaluation set, comprising 495 and 115 tasks, respectively. We employ a high-performing instruction-tuned LLaMA-family model to collect expert trajectories on the training set. For each task, the inference temperature is set to 1, and four trajectories are generated. The trajectory with a final reward of 1 is selected as the expert trajectory; if multiple such trajectories exist, one is chosen at random, and if none achieves a reward of 1, the task is discarded. This process yields expert trajectories for 452 tasks, resulting in a total of 5,239 ⟨observation, action⟩ pairs.

For the world model data, we use the target model to propose five action candidates for each observation in the expert trajectories. To avoid repetitive tool calls and promote exploration, we remove the tool used in the expert action from the corresponding tool set of each expert observation, allowing the model to select from the remaining tools. The selected action is then executed in the environment to obtain the next observation. Each resulting ⟨expert observation, action, next observation⟩ triplet is included in the training dataset for the world model.

For the self-reflection data, for each ⟨expert observation, expert action⟩ pair, we selecte three alternative actions from the five corresponding world model datapoints and present them to the model itself for reflection, prompting it to explain the rationale behind the expert action choice. We filter out a small number of low-quality reflection samples, resulting in a total of 5,233 training instances.

We adopt `LLaMA-Factory` as the training codebase. For imitation learning, we train for 6 epochs with a learning rate of 1e-5. For implicit world model learning, we train for 1 epoch with a learning rate of 5e-6. For self-reflection, we conduct 6 epochs of SFT with a learning rate of 1e-5. In all training configurations, the batch size is fixed at 16. We report the full table in Table 1 as `Tau-Bench` does not contain more fine-grained metrics.

| **Training Example — `Tau-Bench`** | |
|---|---|
| **Task:** | Act as a retail agent to assist customers in modifying their orders. |
| **Current State:** | Sure! My name is Sophia Nguyen, and my zip code is 20171. |
| **Expert Action:** | "function_call": "find_user_id_by_name_zip", "arguments": "first_name": "Sophia", "last_name": "Nguyen", "zip": "20171" |
| **Implicit World Modeling:** | Action: "find_user_id_by_name_and_zip"
Next observation: "Unknown action find_user_id_by_name_and_zip" |
| **Self-Reflection:** | The customer has provided their name, Sophia Nguyen, and their zip code, 20171. The goal here is to identify the user based on these details so we can provide them with the appropriate service or information they're seeking. The candidate action, find_user_id_by_name_zip, seems to be the right approach as it directly uses the name and zip code provided by the customer to locate their user ID. However, the alternative actions listed all have the same name, find_user_id_by_name_and_zip, but they result in an unknown action outcome, indicating that this specific function or command is not recognized within the system. This suggests that these alternatives are not valid options for the current system and cannot be used effectively. Therefore, I would proceed with the candidate action, find_user_id_by_name_zip, because it's the only viable option that matches the customer's provided information and has produced a valid user ID outcome, sophia_nguyen_2370. This action aligns perfectly with the objective of identifying the customer based on their name and zip code. The action is: {'arguments': '{"first_name":"Sophia","last_name":"Nguyen","zip":"20171"}', 'name': 'find_user_id_by_name_zip'} |

## F.5 SEARCHQA

**Expert Trajectories Collection.** We selected all 3-hop and 4-hop tasks from the Musique training dataset, along with 1,438 randomly sampled 2-hop tasks, to fit within scenarios that require multi-step reasoning for solving complex problems. Finally, we have 7,000 tasks in total. Since the training data lacks fine-grained reasoning traces like the thinking–search–answer structure as Jin et al. (2025), we used the Search-R1 model to generate expert data. Specifically, we set the temperature to 1.0 and generated 5 trajectories for each task, retaining only those whose final answers match the ground truth. To reduce redundancy, we keep at most 2 correct trajectories per task. This process yields 2,082 trajectories containing a total of 7,691 state–action pairs for imitation learning.

**World Modeling Data Construction** Consistent with the observations of Jin et al. (2025), we find that directly predicting the content of retrieved documents yields suboptimal performance, as many tokens are not directly relevant to the search query. To address this, we first instruct the model to summarize the retrieved documents, and then let the model predict these summaries rather than the full text. For each state in the expert trajectory, we let the model generate 30 alternative actions with a temperature of 1.0, enabling it to internalize the environment dynamics from its own early experiences substantially. If a generated action is invalid that the query is not enclosed within the ¡search¿¡/search¿ tags, we return the feedback: "Format error! You must enclose the search query within the ¡search¿¡/search¿ tags if external knowledge is required."

**Self-Reflection Data Construction** To construct the self-reflection training dataset, we randomly sample 2 alternative actions for each state. For each instance, the model is prompted to generate fine-grained reasoning explaining why the expert action is superior to the alternatives, based on the current state, the expert action, the alternative actions, and the retrieval documents associated with these actions. This procedure resulted in 7,691 training data containing detailed reasoning processes.

**Training Details** For Supervised Fine-Tuning (SFT), we adopt LLaMA-Factory as the codebase and perform full-parameter tuning on 4 H100 GPUs using ZeRO-3. Specifically, for imitation learning and self-reflection, we train for 3 epochs with a learning rate of $1 \times 10^{-5}$, a context window of 8192 tokens, and a batch size of 2 per GPU, while setting the gradient accumulation steps to 16. For implicit world model learning, we utilize the world model data from expert trajectories to achieve a 1:1 ratio with the imitation learning dataset and train under the same settings.

For Reinforcement Learning (RL), we employ the Search-R1 codebase and train on 8 H100 GPUs. All settings are kept consistent with Jin et al. (2025), except that we use the F1 score as the reward, set the maximum number of retrieval interactions to 6, configure the context window to 12,280 tokens, and specify a maximum output length of 2,048 tokens. For the training data, we use all training tasks from the MuSiQue dataset.

An example illustrating the inputs for different methods is provided below, with the full results presented in Table 7.

| **Training Example — `SearchQA`** | |
|---|---|
| **Task:** | Who was the ninth governor-general in the country where Boridi is located? |
| **Current State:** | `<think>` I need to find the ninth governor-general in the country where Boridi is located. I'll search for it. `<search>` Boridi `<information>` Doc 1 (Title: Boridi) Boridi is a village in the Owen Stanley Range in Central Province, Papua New Guinea. Boridi is a Koiari village . . . |
| **Expert Action:** | `<think>` I found out that Boridi is located in Papua New Guinea. Now I need to find the ninth governor-general in Papua New Guinea. `<search>` ninth governor-general in Papua New Guinea |
| **Implicit World Modeling:** | • `<search>` ninth governor-general in Papua New Guinea → "The ninth Governor-General of Papua New Guinea was Sir Michael Ogio . . ." 
 • `<search>` governor-general of Papua New Guinea history → "The first Governor-General of Papua New Guinea was Sir John Guise . . ." 
 • `<search>` Papua New Guinea governor-general list → "The Governor-General of Papua New Guinea is the representative of the monarch . . ." |
| **Self-Reflection:** | Okay, I've found that Boridi is located in Papua New Guinea. Now, I need to find the ninth governor-general of Papua New Guinea. **I'm going to search for 'ninth governor-general in Papua New Guinea'. This query is specific and should return the information I need.** I'm hoping to find a document that mentions the ninth governor-general by name and provides some information about their background and term in office. I'm also considering searching for 'governor-general of Papua New Guinea history' or 'Papua New Guinea governor-general list', **but I think the first query is more likely to return the information I need.** |

Table 7: Performance on SearchQA.

| Type | Method | SearchQA | | | | |
|---|---|---|---|---|---|---|
| | | Musique | HotpotQA | 2wiki | Bamboogle | All |
| *Base:* `Llama-3.2-3B-Instruct` | | | | | | |
| Prompting | | 13.3 | 24.3 | 22.2 | 35.1 | 21.1 |
| +RL (GRPO) | | 25.0 | 35.0 | 27.8 | 35.4 | 29.7 |
| Imitation Learning | Behavior Cloning | 38.0 | 46.8 | 30.6 | 55.2 | 39.8 |
| + RL (GRPO) | | 43.6 | 47.8 | 38.4 | 63.2 | 44.8 |
| Early Experience | Implicit World Modeling | 39.0 | 49.5 | 37.8 | 59.1 | 43.4 |
| + RL (GRPO) | | 44.5 | 53.1 | 50.1 | 63.6 | 50.3 |
| Early Experience | Self-Reflection | 38.6 | 47.3 | 37.0 | 58.4 | 42.3 |
| + RL (GRPO) | | 42.7 | 50.7 | 43.2 | 57.9 | 46.5 |
| *Base:* `Llama-3.1-8B-Instruct` | | | | | | |
| Prompting | | 21.0 | 39.5 | 31.3 | 49.4 | 32.1 |
| +RL (GRPO) | | 33.1 | 40.1 | 37.9 | 54.4 | 38.4 |
| Imitation Learning | Behavior Cloning | 41.0 | 49.3 | 42.6 | 58.8 | 45.4 |
| + RL (GRPO) | | 47.0 | 51.0 | 40.2 | 59.7 | 47.1 |
| Early Experience | Implicit World Modeling | 44.3 | 53.1 | 43.9 | 58.6 | 48.0 |
| + RL (GRPO) | | 50.6 | 50.8 | 45.6 | 59.7 | 49.8 |
| Early Experience | Self-Reflection | 41.8 | 53.1 | 46.4 | 58.3 | 48.0 |
| + RL (GRPO) | | 47.7 | 52.9 | 50.2 | 59.4 | 51.0 |

## F.6 SCIENCEWORLD

We follow the default split of `ScienceWorld` (Wang et al., 2022) with the `AgentGym` (Xi et al., 2024) setup under the Verl-Agent (Feng et al., 2025) framework. From the expert trajectories in `ScienceWorld`, we extract 14,506 state–action pairs to form $\mathcal{D}_{\text{expert}}$ These expert trajectories are optimal given the completeness of task solvability in the dataset.

For implicit world modeling, we augment $\mathcal{D}_{\text{expert}}$ with $\mathcal{D}_{\text{rollout}}$. At each state, we sample 3 non-expert actions uniformly without replacement from the admissible action list (excluding the expert action) and include the expert action for implicit world modeling.

For self-reflection, we construct data by prompting the model to explain its own decisions. For each state, we use the same policy model with temperature 1.0 to propose up to 3 alternative actions (2 alternative actions for 🦙 `Llama-3.1-8B-Instruct` ). We canonicalize proposed actions and keep only unique ones. If a proposed action is not in the admissible action space for that state, we discard it and instead sample uniformly at random from the remaining unselected admissible actions. The final prompt asks the model to justify why the expert action is preferable to the sampled alternatives given the current state and available tools.

For all the training and evaluations, we use one-shot example, which is shown above. During training, we use a batch size of 32 and a learning rate of $5\mathrm{e}{-6}$, and train with `LlamaFactory` (Zheng et al., 2024b) for 1 epoch. For the evaluation, we set the maximum prompt length to be 4096, the maximum response length to be 1024, and the temperature to be 0.4. Below is a data example illustrating the inputs for different methods given the same task and state (we list three examples for implicit world modeling). We report the full table in Table 1 as `ScienceWorld` does not contain more fine-grained metrics.

| **Training Example — `ScienceWorld`** | |
|---|---|
| **Task:** | Your task is to determine if aluminum foil is electrically conductive. The aluminum foil is located around the living room. First, focus on the aluminum foil. If it is electrically conductive, place it in the green box. If it is electrically nonconductive, place it in the blue box. |
| **Current State:** | This room is called the workshop. In it, you see: the agent a substance called air a table. On the table is: a battery, a black wire, a green light bulb, which is off, a orange light bulb, which is off, a orange wire, a red light bulb, which is off, a red wire, a switch, which is off. a ultra low temperature freezer. The ultra low temperature freezer door is closed. You also see: A door to the hallway (that is closed). |
| **Expert Action:** | connect battery cathode to yellow wire terminal 1 |
| **Implicit World Modeling:** | • connect inventory to orange → "terminal 1 on inventory is now connected to terminal 1 on orange."
• connect black wire terminal 2 to anode in green light bulb → "terminal 2 on black wire is now connected to anode on green light bulb."
• connect cathode in orange light bulb to freezer → "cathode on orange light bulb is now connected to terminal 1 on ultra low temperature freezer" |
| **Self-Reflection:** | I'm currently in the workshop, and my task is to determine if the aluminum foil is electrically conductive. To do this, I need to place it in either the green box or the blue box, depending on its conductivity. I've recently moved the aluminum foil to the workshop, and now I see it sitting here, along with various other objects like the electric motor, the ultra low temperature freezer, and a bunch of wires and light bulbs. Looking at my recent actions, I connected the anode of the battery to terminal 1 on the green wire. This was a deliberate step to set up a circuit, but I'm not sure what the ultimate goal is yet. I also see that I have the option to connect the cathode of the battery to terminal 1 on the yellow wire, which would complete the circuit. However, I'm not sure if this is relevant to determining the conductivity of the aluminum foil. I also notice that I have the option to connect the door to the ultra low temperature freezer, which would result in the door being connected to the ultra low temperature freezer. This seems like a strange action, and I'm not sure what the purpose of this would be. It's possible that this action is related to the workshop's equipment, but it doesn't seem directly relevant to my task. Another option I have is to move the yellow light bulb to the blue box. This would simply move the light bulb to a different location, but it doesn't seem to have any bearing on my task. Considering my goal of determining the conductivity of the aluminum foil, I need to think about how I can use the available equipment to test its conductivity. *I'm not sure what the properties of the aluminum foil are, but I know that if it's conductive, it should allow electricity to flow through it.* Given this, **I think the most suitable action would be to connect the cathode of the battery to terminal 1 on the yellow wire**. This would complete the circuit and allow me to test the conductivity of the aluminum foil. If the aluminum foil is conductive, it should allow electricity to flow through it, and I can place it in the green box. If it's not conductive, it won't allow electricity to flow, and I can place it in the blue box. This action seems like the most logical choice because it directly addresses my task and allows me to test the conductivity of the aluminum foil. The other actions I have available don't seem to be directly relevant to my task, and they might even be distracting or confusing. By connecting the cathode of the battery to terminal 1 on the yellow wire, I can focus on my task and make progress towards determining the conductivity of the aluminum foil. |

## F.7 TRAVELPLANNER

We adapt the `TravelPlanner` (Xie et al., 2024a) benchmark into a gym-based environment for training language agents. The original benchmark contains 1,225 queries divided into training (45 queries), validation (180 queries), and test sets. We use the 45 training trajectories that cover diverse planning scenarios across different difficulty levels based on trip duration (3, 5, or 7 days) and constraint complexity (Easy, Medium, Hard). *Easy* queries are primarily budget-constrained for a single person. *Medium* queries introduce additional constraints such as cuisine type, room type, or room rules, with the number of travelers varying between 2 and 8. *Hard* queries include transportation preferences along with all medium-level constraints, containing three randomly selected hard constraints. We evaluate on the validation set of 180 queries.

**Environment Implementation.** We implement TravelPlanner as a gym environment with discrete action spaces and dictionary observation spaces. The state representation includes the current planning progress formatted in a structured text format: query description, budget tracking (initial/spent/remaining), and the current plan status for each day showing transportation, meals, attractions, and accommodation fields. Actions are JSON objects with fields for action type (*e.g.*, SET_TRANSPORTATION, SET_MEAL, SET_ACCOMMODATION), day number, field name, selected value, and cost. The action space dynamically generates all valid actions based on available data from reference information, including flights between cities, restaurants with cuisine types and prices, attractions, and accommodations with room rules and minimum night requirements. The environment tracks budget spending, validates constraints in real-time, and maintains planning progress through a state machine that advances through each field sequentially.

**Expert Trajectory Collection.** We use 45 annotated trajectories from the training set as expert demonstrations $\mathcal{D}_{\text{expert}}$. Each trajectory contains a complete multi-day travel plan with ground-truth actions for transportation, accommodation, dining, and attractions. We decompose these trajectories into 1,395 individual state-action pairs using the SFTConverter, which maps expert plan entries to valid gym actions while handling city name variations and validating against environment constraints.

**Implicit World Modeling.** For world modeling data, we generate two types of training examples. First, we reformat the expert trajectories into state-transition format where the model learns to predict next states given current state and action. Second, we perform exhaustive augmentation by executing ALL available valid actions at each state in the expert trajectories (not just sampling), collecting comprehensive state transitions to maximize coverage of environment dynamics. This process generates over 70,000 state-transition samples, providing rich supervision for learning environment dynamics including budget updates, constraint evaluations, and plan progression.

**Self-Reflection.** We construct self-reflection data by prompting Llama-3.1-8B-Instruct to generate chain-of-thought reasoning explaining why expert actions are preferable to alternatives. For each of the 1,395 state-action pairs, we explore up to 30 alternative valid actions and generate reasoning that considers multiple constraints: budget limits, minimum night stays for accommodations, restaurant diversity requirements, and round-trip completion. The reasoning generation uses temperature 0.9 with 8-way tensor parallelism to produce natural explanations while maintaining logical consistency. We do not apply additional filtering as the reasoning generation already validates constraint satisfaction.

**Training Details.** We train models using `LlamaFactory` with full fine-tuning on 8 H100 GPUs using DeepSpeed ZeRO-3. For imitation learning and implicit world modeling, we train for 5 epochs with learning rate $1e-5$ and cosine scheduler. For self-reflection, we extend the maximum generation length to 8K tokens to accommodate detailed reasoning. All models use 32K context windows with batch size 16 per GPU. For evaluation, we use vLLM with tensor parallelism across 8 GPUs and greedy decoding to ensure reproducibility.

A data example showing inputs for different methods under the same task and state is given below. Full results are in Table 8.

| Training Example — `TravelPlanner` | |
|---|---|
| **Task:** | Please help me plan a trip from St. Petersburg to Rockford spanning 3 days from March 16th to March 18th, 2022. The travel should be planned for a single person with a budget of $1,700. |
| **Current State:** | Total Days: 3, Initial Budget: $1700, Spent: $0, Remaining: $1700 Day 1: transportation: PENDING, breakfast: PENDING, attraction: PENDING, lunch: PENDING, dinner: PENDING, accommodation: PENDING Next action required: Plan day 1 transportation Available: Flight F3573659: $474, Departure 15:40, Arrival 17:04 |
| **Expert Action:** | {"action_type": "SET_TRANSPORTATION", "day": 1, "field": "transportation", "value": "Flight Number: F3573659, from St. Petersburg to Rockford, Departure Time: 15:40, Arrival Time: 17:04", "cost": 474} |
| **Implicit World Modeling:** | • SET_TRANSPORTATION (F3573659) → "After this action, you've spent $474, leaving $1226 from your $1700 budget (28% used). Good start! You're traveling to your destination." 
 • SKIP_TRANSPORTATION → "After this action, you've spent $0, leaving $1700 from your $1700 budget (0% used)." |
| **Self-Reflection:** | Important: Consider all constraints. Flight F3573659 is a valid option at $474. It satisfies all constraints and gets me to my destination efficiently. The only alternative is skipping transportation, which prevents reaching the destination. Given all constraints and optimization factors, Flight F3573659 is indeed the best choice as it's the most cost-effective option that enables the trip. **Therefore, Flight F3573659 is optimal.** |

Table 8: Performance on TravelPlanner with different training approaches.

| Model | Method | Commonsense Constraint | | Hard Constraint | | Final |
|---|---|---|---|---|---|---|
| | | Micro | Macro | Micro | Macro | Pass Rate |
| *Base:* Llama-3.1-8B | | | | | | |
| Prompting | | 36.9 | 0.0 | 61.0 | 46.7 | 0.0 |
| Imitation Learning | Behavior Cloning | 82.6 | 25.0 | 54.8 | 46.7 | 17.2 |
| Early Experience | Implicit World Modeling | 84.0 | 38.9 | 56.9 | 42.2 | 25.0 |
| Early Experience | Self-Reflection | 84.7 | 42.2 | 61.0 | 51.1 | 32.2 |
| *Base:* Llama-3.2-3B-Instruct | | | | | | |
| Prompting | | 34.2 | 0.0 | 58.1 | 48.9 | 0.0 |
| Imitation Learning | Behavior Cloning | 81.7 | 27.8 | 57.1 | 46.1 | 19.4 |
| Early Experience | Implicit World Modeling | 84.7 | 44.4 | 56.7 | 42.8 | 28.3 |
| Early Experience | Self-Reflection | 87.0 | 46.1 | 61.9 | 52.2 | 32.2 |

## F.8 WEBARENA

Given that the full evaluation set in `WebArena` (Zhou et al., 2024) is lengthy and includes many similar tasks, we follow prior work (Qi et al., 2024; Wei et al., 2025b) and evaluate our trained agents on `WebArena-Lite` (Liu et al., 2024), a more efficient and balanced subset of 165 high-quality, challenging tasks, hand-selected from the original 812. Therefore, the remaining 647 tasks in WebArena, excluding those in the evaluation set, are used for agent training.

To obtain expert demonstrations in `WebArena`, we extract successful trajectories from the highest-performing agents on the public WebArena leaderboard.[1] Specifically, we select those that include accessibility tree information in their observations, such as IBM CUGA (Marreed et al., 2025), ScribeAgent (Shen et al., 2024), Learn-by-Interact (Su et al., 2025), and AgentOccam (Yang et al., 2024). After filtering out the unsuccessful trajectories, we obtain 554 successful ones and 7,044 state-action pairs, forming $\mathcal{D}_{\text{expert}}$.

To branch out from the expert trajectories for implicit world modeling, we augment $\mathcal{D}_{\text{expert}}$ to form $\mathcal{D}_{\text{rollout}}$. For each state in $\mathcal{D}_{\text{expert}}$, we let the target model (to be trained) propose 5 non-expert actions using free-form generation, excluding any that are identical to the expert action. For each resulting next state, we apply an additional processing step: using the same model, we generate a concise summary of the next-state observation conditioned on the task, replacing the raw observation to reduce noise and emphasize task-relevant information. We then include the expert action together with the sampled ones to create triplets of the form (current state, action, summarized next state), resulting in $7,044 \times 6 = 42,264$ triplets in total for each model.

For self-reflection, we construct $\mathcal{D}_{\text{SR}}$ by prompting the model to explain why the expert action is preferable to the sampled alternatives in the current state. We use the same 5 alternatives from $\mathcal{D}_{\text{rollout}}$, canonicalize action strings to avoid duplicates, and replace any invalid actions (*e.g.*, referring to non-existent UI elements) with randomly sampled valid ones. The final prompt includes the current state, the admissible actions, and the expert action, and asks the model to justify the optimality of the expert choice in terms of task progress, constraint satisfaction, and efficiency. We filter out low-quality generations where the explanation incorrectly supports a non-expert action, leaving 3,190 high-quality self-reflection examples.

All models are trained for 2 epochs with learning rate $1\mathrm{e}-5$ and cosine scheduler. We report our full numbers on `WebArena-Lite` in Table 9.

---

[1] `https://docs.google.com/spreadsheets/d/1M801lEpBbKSNwP-vDBkC_pF7LdyGU1f_ufZb_NWNBZQ/edit?usp=sharing`

**Training Example — `WebArena`**

| | |
|---|---|
| **Task:** | What are the top-3 best-selling products in January 2023? |
| **Current State:** | [1281] RootWebArea `'Products / Inventory / Catalog / Magento Admin'`
  [1334] link `'Magento Admin Panel'`
    [1343] img `'Magento Admin Panel'`
  [1286] menubar `''` orientation: horizontal
    [1288] link `'ue604 DASHBOARD'`
    [1292] link `'ue60b SALES'`
    [1296] link `'ue608 CATALOG'`
    [1300] link `'ue603 CUSTOMERS'`
    [1304] link `'ue609 MARKETING'`
    [1308] link `'ue602 CONTENT'`
    [1312] link `'ue60a REPORTS'`
...
    [2146] button `'ue605Filters'`
... |
| **Expert Action:** | Click `[1312]`. |
| **Implicit World Modeling:** | • click [1312] → The user clicked on the "REPORTS" link in the horizontal menubar, which changed the focus to this link. A new vertical menu expanded below it, showing categories such as Marketing, Reviews, Sales, Customers, Products, Statistics, and Business Intelligence, each with sub-links. The main product table remains visible, but the navigation context has shifted to the reports section, enabling access to detailed reporting options.
• click [2146] → Clicking the "Filters" button shifted focus to the "Search by keyword" textbox and likely opened the filter/search area. The product list updated to show different products such as "Sol Active Short-32-Green" and "Sol Active Short-33-Purple", indicating that the action updated the product list via filtering.
• go_back → Navigates from the "Products / Inventory / Catalog" page to the "Dashboard / Magento Admin" page. The view changes from a product table to a dashboard with summary information such as revenue, tax, shipping, and tabs like Bestsellers and Most Viewed Products. This navigation may reset the page state. |
| **Self-Reflection:** | The goal is to find the top-3 best-selling products in January 2023. The current "Products / Inventory / Catalog" page lists inventory details but not sales data or best-seller rankings. Clicking on "REPORTS" is the most logical choice, as it typically contains sales metrics, customer insights, and product performance reports. From there, it should be possible to select sales reports, filter by January 2023, and retrieve the ranked list of best sellers. Alternatives such as filtering or searching products on the catalog page will not reveal sales statistics, and going back to the dashboard may only provide limited summary widgets without detailed filtering options. **Therefore, selecting "REPORTS" directly aligns with the task objective** and is the optimal next step to access the necessary data. |

Table 9: Performance on `WebArena-Lite`. Results of closed-source models are borrowed from Qi et al. (2025).

| Type | Method | WebArena-Lite | | | | | |
| --- | --- | --- | --- | --- | --- | --- | --- |
| | | Reddit | Gitlab | CMS | Map | OSS | Avg. SR |
| *Base: Closed-Source Model* | | | | | | | |
| Prompting | GPT-4-Turbo | 10.5 | 16.7 | 14.3 | 36.7 | 13.3 | 17.6 |
| Prompting | GPT-4o | 10.5 | 10.0 | 20.0 | 20.0 | 11.1 | 13.9 |
| *Base:* Llama-3.2-3B-Instruct | | | | | | | |
| Prompting | | 0.0 | 0.0 | 0.0 | 1.7 | 3.2 | 1.2 |
| Imitation Learning | Behavior Cloning | 0.0 | 7.7 | 10.3 | 0.0 | 3.2 | 6.1 |
| Early Experience | Implicit World Modeling | 0.0 | 15.4 | 2.4 | 13.8 | 9.7 | 8.5 |
| Early Experience | Self-Reflection | 11.1 | 15.4 | 11.9 | 3.5 | 6.5 | 7.3 |
| *Base:* Llama-3.1-8B-Instruct | | | | | | | |
| Prompting | | 0.0 | 0.0 | 7.1 | 0.0 | 0.0 | 0.6 |
| Imitation Learning | Behavior Cloning | 0.0 | 0.0 | 0.0 | 11.9 | 8.0 | 4.9 |
| Early Experience | Implicit World Modeling | 11.1 | 0.0 | 7.3 | 8.6 | 16.1 | 8.5 |
| Early Experience | Self-Reflection | 0.0 | 15.4 | 11.9 | 10.3 | 3.2 | 8.5 |
| *Base:* Llama-3.3-70B-Instruct | | | | | | | |
| Prompting | | 11.1 | 15.4 | 17.1 | 7.1 | 3.2 | 9.1 |
| Imitation Learning | Behavior Cloning | 0.0 | 8.3 | 14.3 | 17.2 | 16.1 | 13.3 |
| Early Experience | Implicit World Modeling | 8.3 | 16.7 | 16.7 | 19.0 | 19.4 | 16.4 |
| Early Experience | Self-Reflection | 0.0 | 16.7 | 23.8 | 14.0 | 19.4 | 15.2 |

