# OpenReview forum: "Agent Learning via Early Experience"
_ICLR.cc/2026/Conference — Submitted to ICLR 2026_

### Official Review · Reviewer_b21C · 2025-10-20

**Soundness:** 3
**Presentation:** 3
**Contribution:** 2
**Rating:** 4
**Confidence:** 4

**Summary:**

This paper points out two limitations of using fixed offline datasets and collecting data (including reward signals) through interaction with the environment. It proposes two approaches for training language agents to learn and improve from their own experiences without relying on external reward signals: (1) implicitly training a world model that predicts the next state given the current state and a non-expert alternative action generated by the model, and (2) predicting the rationale for comparing the expert action with the alternative action. The authors evaluate two language models using their proposed method across eight diverse environments.

**Strengths:**

This work introduces the concept of early experience, in which interaction data generated by the agent’s own actions provide supervision through the resulting future states, without relying on reward signals. Two simple supervision strategies are proposed, which could broaden the potential applications of language agents.

**Weaknesses:**

To apply the proposed strategies, the environment dynamics need to be accessible in order to efficiently collect early experience as the behavioral agent acts in arbitrary states. This requirement may limit the range of environments to which these approaches can be applied. The paper could benefit from examples of reward-free environments where designing a dense reward function is particularly challenging. Consider moving the brief explanation of the environments to the beginning of the experiment section for improved clarity and context.

**Questions:**

(1) How do the next states $s^j_i$ encode implicit feedback about action quality? Since all alternative actions differ from those in the offline dataset, how can action quality be evaluated without a reward signal, based solely on the resulting next state? Similarly, $c^j_i$ appears to represent a form of explicit feedback that functions analogously to reward signals in the Self-Reflection process.

(2) In the example in Section 4.3, the reasoning appears to be partially hallucinated. Could you comment on this observation?

(3) Please consider evaluating with additional baselines to better isolate the effect of each component:

- Data augmentation via alternative actions without IWM. How impactful is the inclusion of IWM?
- IWM trained with only the offline dataset.
- Reinforcement learning with LLM-assigned numerical rewards.
- Unsupervised RL methods.

(4) Please include an evaluation using a larger model (in Section 6.2).

(5) Could you provide examples of $a^j_i$ and corresponding $s^j_i$, and compare them with those of the expert?

**Minors**

(6) When applying your approaches, do you combine each proposed loss with the imitation learning loss?

(7) Which datasets are used for training and evaluation?

(8) How are the experimental results are statistically consistent?

---

> ### Author Response · Authors · 2025-11-21
> **Response to Reviewer b21C**
>
> Thanks for reviewing our submission. We address several shared concerns in the General Response, and provide reviewer-specific replies below.
>
> ### 1: Environment Must Be Accessible
> Our method requires that the environment can return the next state after an action. This matches the deployment setting of modern LLM-based agents in practice. Web automation, multi-step tool use, travel planning, embodied simulation, and command-line tool environments all expose state transitions naturally through page loads, API calls or tool outputs.
> The eight environments we evaluate cover embodied simulation, travel planning, shopping tasks, web navigation, terminal tool use, and tool-agent-user interactions. They are comprehensive and representative of real LLM-agent applications where obtaining next states is straightforward. If there are practical environments where this assumption does not hold, we would appreciate concrete examples. That would help us better understand the reviewer’s concern and clarify the boundaries of applicability.
>
>
> ### 2: Adding Reward-Free Environments
> As described in the abstract and introduction, many real-world LLM-agent settings do not provide dense or reliable reward signals. Examples include web automation and multi-turn tool-use tasks where designing consistent rewards is difficult or impractical. This is precisely why we *have already included such environments in our evaluation*.
>
> Among the eight benchmarks, five do not provide dense rewards. Tasks like Tau-Bench, WebArena, TravelPlanner, and several others in our study all fall into this reward-free category, and our method is shown to work effectively across them. If the reviewer has specific environments in mind, we would greatly appreciate concrete examples so that we can better understand the concern and clarify applicability in future work.
>
>
> ### 3: Explain Environments
> Thank you for the helpful suggestion. Please refer to the General Response 2. We will revise it in the next version.
>
>
> ### Q1. Implicit Feedback About Action Quality
> For IWM, we never infer action quality directly from the next state. The next-state prediction is an auxiliary grounding signal, not a reward surrogate. Action quality is essentially supervised by the expert action, rather than the environment or the model.
> Next-state prediction improves **environment understanding**, not action scoring.
>
>
>
> ### Q2. CoT Hallucination
> As shown in Appendix E.1, we give both the expert action and the policy-proposed action to the same instruction-tuned model to simply let the model explain why the expert action is potentially better.
> This is a “given-answer-then-explanation” setting rather than genuine multi-step reasoning. Prior work shows that such explanations are low-effort for LLMs and do not require faithfulness or deep planning [1–3].
> In addition, models still need to output future states during training, which still gives meaningful supervision signals; we provide examples in each environment in Appendix F.
>
> [1] Turpin et al., Language Models Don’t Always Say What They Think: Unfaithful Explanations in Chain-of-Thought Prompting, NeurIPS 2023.
>
> [2] Paul et al., Making Reasoning Matter: Measuring and Improving Faithfulness of Chain-of-Thought Reasoning, EMNLP 2024.
>
> [3] Qu et al., ReCoT: Reflective Self-Correction Training for Mitigating Confirmation Bias in Large Vision-Language Models, ICCV 2025.
>
>
>
> ### Q3. Additional Baselines
> We include additional baselines and comparisons in Appendix D.1 due to space constraints, and will move them into the main paper if space permits.
>
> ### Q4. Larger Models
> Model scaling results are provided in Appendix D.2 and General Response #4. We will relocate them to the main paper if space allows.
>
> ### Q5 and 7. Examples and Environments Used in Training and Evaluation
> Concrete examples and environment-specific training/evaluation details are provided in Appendix F. General experiment setup is provided in Appendix E.2.
>
> ### Q6. Combination of Imitation Learning
> Yes. The losses are combined with imitation learning, as described in Lines 258–260 and 283–286.
>
> ### Q8. Experiment Significance
> Due to limited compute and the breadth of environments evaluated, we do not run multiple independent seeds. However, we observe that the methods transfer reliably across environments and model scales without doing environment-specific adjustment once the training recipe is fixed. We will include CIs where feasible in future revisions.

---

### Official Review · Reviewer_LDNN · 2025-10-30

**Soundness:** 3
**Presentation:** 3
**Contribution:** 3
**Rating:** 6
**Confidence:** 4

**Summary:**

This paper focuses on the goal of training language agents to accomplish tasks. They identify that supervised fine-tuning is a difficult paradigm to scale in many settings. Instead, they propose what they call a middle ground between IL and RL that uses reward free data (produced by environment interaction) to improve the agent. The two methods they propose are using the data for world modeling with the LLM and "self-reflection" where the LLM thinks about an the actions taken and explains the rollout to generate more data and improve thinking about a task. The results show their fine tuning is able to achieve superior results to the bast models on a set of agentic LLM tasks. They also explore a set of generalization ablations on OOD tests.

**Strengths:**

The paper introduces an interesting paradigm for bridging supervised learning setups and reinforcement learning setups. This problem is well motivated and they provide some interesting specific ways to approach this paradigm.

The empirical results on their method for augmenting the dataset are strong.

The paper is generally well written and clear.

**Weaknesses:**

Need to have CIs on all the plots and table. As well the plots at 396 could probably be a table? While the plots are pretty, I think doing it this way and shortening the y axis isn't the best way to do it.

I wish you had a limitations in your discussion. In my view a large limitation here is the ability to handle longer horizon tasks. If you are just doing a single action at states that is fine, but if something takes two actions then your model will never learn it. As well, you need a model or simulation of the world right? Without that you can't reset the states for your method. As well, since we are querying an already fine-tuned model we are limited to settings in which we are already moderately performant (I think?).

I'll put some questions below that I view are unclear parts of the paper that hopefully could be addressed. I'll gladly raise my score if they are addressed or I am mistaken.

**Questions:**

Around 219 it is described how the dataset is created. Is only one step appended? So is this a recursive-type generation process where states are added, then we redo it on new states? Or is this a single-pass setup?

Around line 251 you say you're using the same model parameters for both world modeling and policy modeling. I don't know if I believe this claim the model internalizes the dynamics implicitly. How is it done? Do we have multiple heads on the same transformer backbone? I would almost like to see just training on random states and actions as an ablation to see if the true dynamics makes any difference here or if any better initialization would work.

Around 276 you're saying that you train the model to produce chain of thought thinking. I don't quite understand this setup I don't think. So is the model producing some text in "hindsight" and then we are training the model to predict this in a forward fashion? Would it be possible to achieve something similar to this with just prompting? Like if we just gave options and we said "consider these options and choose the best one"? Or would that achieve something different than the optimization here?

You need a world model for your method right? So you can easily query next states from state actions to add to your dataset?

In the 396 figure which checkpoints are these trained from (like which environment are they trained on or is it just the base llama)? To be clear here as well, this is just base GRPO from your checkpoints right? Or am I misunderstanding?

How much data did you produce for training with your early experience methods? If see you have some information in the ablation about this but for your methods it would be nice to know how much data you're producing.

---

> ### Author Response · Authors · 2025-11-20
> **Response to Reviewer LDNN**
>
> Thanks for reviewing our work carefully and getting back to us with valuable comments and interesting questions. We address several shared concerns in the General Response, and provide reviewer-specific replies below.
>
> ### 1. Presentation, CIs, etc.
>
> We agree that reporting CIs improves statistical clarity. Computing CIs across all environments requires rerunning many rollouts, which exceeded our compute budget during the initial submission. We are generating additional runs for several environments and will include CIs in the next revision.
> Due to page limits, we present the plots at line 396 as a figure and place the full results as a table in Appendix F.
>
> ### 2. Limitations
> We include our discussion of limitations in Appendix B due to space constraints. Regarding environment simulation: having access to an environment does not imply we already have performant agents (e.g., web environments remain challenging). The “world models” mentioned in the paper are never used to drive the policy; they only serve as a foundation signal during training. This is why we refer to them as implicit world models.
>
>
> ### Q1. Dataset Creation
>
> Yes, only one step is appended per iteration. To reach deeper states, we parallelize 𝐾+1
> environments from the same initial state: the model samples 𝐾 actions, and we also roll out the expert action. After one step, all environments are synced to the expert state, and we repeat until the expert reaches the goal. This allows the policy to propose actions in deep states that would otherwise be unreachable. In environments like WebArena, with hundreds of possible actions and many “no-op” parses, the initial policy is too weak to explore meaningful states without leveraging expert trajectories.
>
> ### Q2. Shared Model Parameters
> We start from the same base LLM (e.g., Llama-3.1-8B). For implicit world modeling, we use the two-stage training described in Lines 258–260 and Figure 2. For self-reflection, the same model reflects on why its sampled action may be suboptimal compared to the expert action; this reflection text is later used as supervision, detailed in L283-286.
> I am very happy you mentioned “multiple heads”: this is a key difference between prior agents and LLM-based agents. Because LLMs express all outputs as language, both next-state descriptions and actions use the same decoder. Therefore, we do not introduce extra parameters.
>
> ### Q3. Chain-of-Thought Reflection
> Encouraging longer CoT before acting is indeed interesting, but this is generally known as test-time scaling [1], which has been shown to be ineffective for LLM agents [2]. The bottleneck in agent performance is not reasoning depth/length, instead, it is understanding the environment. Our reflection setup trains on hindsight explanations, not forward reasoning. We compare prompting-based and ungrounded data-synthesis baselines in Appendix D.1.
>
> [1] Snell et al., Scaling LLM Test-Time Compute Optimally can be More Effective than Scaling Model Parameters, 2025.
>
> [2] Shen et al., Thinking vs. Doing: Agents that Reason by Scaling Test-Time Interaction, 2025.
>
> ### Q4. Need for a World Model
> No, we don’t need a world model, actual environments are already there. Our vision/motivation on the current tasks of interest like multi-turn tool use and web automation is that we have environments and we can do large-scale rollouts but it’s hard to do RL training. So we don't really need a world model to be sample-efficient, instead, we need to improve policy’s ceiling with native data as much as possible. The engineering effort is briefly mentioned in each environment and the answer to Q1.
>
> ### Q5. Checkpoints in Figure 3
> Each grouped bar contains three checkpoints originating from the same instruction-tuned base model (e.g., Llama-3.2-3B):
>
> The grey one is the Llama-> imitation learning -> RL
>
> The red one is the Llama-> IWM -> RL
>
> The blue one is the Llama-> SR -> RL
>
> RL hyperparameters are identical; only the checkpoint differs (after IL, IWM, or SR).
>
> We have included direct Llama → RL results in Appendix F, and they are lower than any of the three starting checkpoints used in Figure 3. We will clarify this in the next revision.
>
> ### Q6. Data Produced
> All environment-specific data statistics are listed in Appendix F. We will improve cross-referencing between the main text and the appendix.
>
> Thanks again for the thoughtful questions. Please let us know if any additional clarification is needed.

---

### Official Review · Reviewer_sfTJ · 2025-10-31

**Soundness:** 3
**Presentation:** 4
**Contribution:** 3
**Rating:** 6
**Confidence:** 2

**Summary:**

The paper proposes a method called *early experience* to make language agents learn from their own exploration beyond imitating expert demonstrations.  The authors prompt the agent to predict the next state based on the current state and explorational action, and to predict the expert action as well as a sentence indicating why the action should be taken.

**Strengths:**

- The paper is well-structured. The motivation and method description are very clear.
- The proposed method is simple and practicable.

**Weaknesses:**

While the experiment results show that the method can enhance the original language agent that has not been updated during its rollouts,  there seems to be no comparison between the proposed *early experience* and other online-updating techniques. Consequently, it is unclear whether the main contribution lies in a genuinely novel approach or primarily in a combination of existing methods.

**Questions:**

The effectiveness of *self-reflection* relies heavily on external LLM feedback. What will happen if we directly turn this external LLM into an agent through prompting? On the other hand, how to get proper feedback when the agent's rollout is OOD for the LLM?

---

> ### Author Response · Authors · 2025-11-20
> **Response to Reviewer sfTJ**
>
> Thank you for reviewing our submission. We address several shared concerns and clarify the novelty of our approach in the General Response, and provide reviewer-specific replies below.
>
> ### 1. Online-Update Techniques
> Thanks for the question. Online-update techniques such as PPO, GRPO, and related methods typically rely on reward signals, discriminator feedback, or verifier models. In contrast, our method does *not* require any reward directly from the environment, making early-experience training fundamentally different from these online-update approaches.
> Furthermore, Figure 3 shows that our method produces stronger starting checkpoints for online-update techniques such as RL. This indicates that early experience is *not* an alternative to online-updating, but rather *complementary*.
> Our method currently focuses on the offline regime, with its online extension discussed as future work in Appendix B.
> Regarding comparisons with baselines, we report results against other offline methods in Appendix D.1 and clarify the distinction between offline and online techniques in Appendix C.1. We will move this summary into the main paper if space permits.
>
> To further address this concern, we ran additional experiments with DPO [1], which also explores model-generated alternative actions during training (i.e., more “online-like” but still without explicit rewards). The training setup treats policy-generated actions as negative examples whenever they deviate from the expert trajectory. We tune the ratio of negative actions and report the best-performing configuration below:
>
> | Method                 | WebShop | ALFWorld |
> |------------------------|---------|----------|
> | Prompt                | 0.0     | 25.0     |
> | &nbsp;&nbsp; + Long CoT | 1.6     | 28.4     |
> | Imitation Learning    | 47.3    | 80.5     |
> | &nbsp;&nbsp; + Long CoT | 0.0     | 25.8     |
> | &nbsp;&nbsp; + DPO      | 53.1    | 82.8     |
> | &nbsp;&nbsp; + Star     | 25.0    | 74.2     |
> | Ours-IWM             | 58.6    | 85.9     |
> | Ours-SR              | 59.4    | 85.2     |
>
> These results show that both of our early-experience variants remain among the top-performing methods.
> Also, DPO is unstable during training: when training continues beyond ~20 steps, performance often collapses and can even drop to zero.
> This instability highlights the robustness of our early-experience approach: despite using similarly non-expert, model-generated actions, our method consistently improves performance without reward signals.
>
>
> [1] Rafailov et al., Direct Preference Optimization: Your Language Model is Secretly a Reward Model, NeurIPS 2023.
>
>
>
> ### Q. External LLM Feedback
> The external LLM used for self-reflection is the **same off-the-shelf** model as the starting checkpoint of the policy. It is not a larger or more capable model. So we are already “turning this external LLM into an agent through prompting”.
> In terms of the feedback, the reflection prompt (Appendix E.1) shows both the expert action and the policy-proposed action to the model and simply asks it to explain why the expert action is potentially better.
> This is a “given-answer-then-explanation” setting rather than genuine multi-step reasoning. Prior work shows that such explanations are low-effort for LLMs and do not require faithfulness or deep planning [1–3].
>
> Regarding OOD rollouts, our design mitigates the issue by **grounding every reflection in the expert action**: even if the policy’s alternative action is OOD, the model is asked to compare it to a known expert action and explain why the expert action is potentially better. Because the explanation is always grounded to the expert trajectory, the feedback quality remains stable even when the policy’s rollout contains OOD behaviors.
>
> [1] Turpin et al., Language Models Don’t Always Say What They Think: Unfaithful Explanations in Chain-of-Thought Prompting, NeurIPS 2023
>
> [2] Paul et al., Making Reasoning Matter: Measuring and Improving Faithfulness of Chain-of-Thought Reasoning, EMNLP 2024.
>
> [3] Qu et al., ReCoT: Reflective Self-Correction Training for Mitigating Confirmation Bias in Large Vision-Language Models, ICCV 2025.

---

### Official Review · Reviewer_MfPf · 2025-11-01

**Soundness:** 3
**Presentation:** 3
**Contribution:** 2
**Rating:** 2
**Confidence:** 4

**Summary:**

In this work authors improve LLM policy in an imitation learning setup, coming up with implicit world model and self-reflection modules. Important motivation of the authors work is to work without access to environment reward signal (i.e. verifiable rewards).

**Strengths:**

- Perfectly reasonable idea that improves LLM policy.

**Weaknesses:**

- Ultimately, I feel that lack of novelty is a most serious problem of this paper. Idea is very straight forward, even though it is perfectly reasonable. Theoretical take on limitations would improve the presentation in the novelty space.
- Introduction has a lot of repetition. Main points of world modeling and self-reflection were repeated many times. More tighter narrative would help the paper.
- In 3. MDP is defined, authors should explain why in their setup MDP is a reasonable fit. MDP is not reasonable fit in much more simpler environments, such as Minecraft (see for example: Ville Tanskanen, Arto Klami and Ville Hautamäki, On the Importance of Representation in Imitating Human-like Gameplay, CoG 2025.). To me it is unreasonable to state that MDP would be suitable.
- Section 3.1 describes behavioral cloning (BC). It is well known work really badly when observed states were not seen in expert demos. Covariate shift will also mess it up (https://arxiv.org/abs/2102.02872). If offline IL is needed then authors should look for some form of regularized BC, such as regularized via Laplacian.
- I dont understand that if environment is available, such as in the website booking example, then why not to utilize it to the fullest? Adversarial imitation learning schemes can be very helpful in this respect.

**Questions:**

- Using (3) how long trajectories could the resulting world model plausibly predict? Authors could take a look at models such as PlaNet to see state only transitions and action conditioned state transition dynamics.

---

> ### Author Response · Authors · 2025-11-20
> **Response to Reviewer MfPf (1/3)**
>
> Thanks for reviewing our submission. We address several shared concerns in the General Response, and provide reviewer-specific replies below.
>
> ### 1. Novelty
> Granted there are previous works, *prior to the LLM era*, that use world models for policy improvement. But how to improve LLM-based policy learning requires novel designs considering the speciality of LLM: the data needed, the environment that previous small neural policies never touched, and the new supervision format (e.g., next-token prediction).
> Please refer to the General Response #1 for more details.
>
> Because of the strong potential of language agents, recent works have been developing language agents for real-world tasks, including explicit external world models for model-based planning [1,2,3] and self-reflection on the reasoning and reward-available settings [4,5] .
> To the best of knowledge, we haven’t noticed any work using implicit world modeling or self-reflection for language agents yet. We have clarified the difference between our work and prior works on Appendix C.1 and we will include it if space allows.
>
> [1] Hao et al., Reasoning with Language Model is Planning with World Model, EMNLP 2023.
>
> [2] Chae et al., Web Agents with World Models: Learning and Leveraging Environment Dynamics in Web Navigation, ICLR 2025.
>
> [3] Gu et al., Is Your LLM Secretly a World Model of the Internet? Model-Based Planning for Web Agents, TMLR 2025.
>
> [4] Madaan et al., Self-refine: Iterative refinement with self-feedback, NeurIPS 2023.
>
> [5] Huang et al., Large Language Models Cannot Self-Correct Reasoning Yet, ICLR 2024.
>
>
>
> ### 2. Introduction Repetitiveness
> Thanks for sharing with us your opinion. We will identify the repetitions and revise correspondingly. We see this as a minor issue, and if you have more specific suggestions, feel free to follow up.

---

> ### Author Response · Authors · 2025-11-20
> **Response to Reviewer MfPf (2/3)**
>
> ### 3. MDP Formulation
> #### 3.1 What is MDP?
> The concept of Markov Decision Process (MDP) originates with reinforcement learning itself. In fact, In [Sutton and Barto], “a reinforcement learning task that satisfies the Markov property is called a Markov decision process (Sec. 3.6)”. In our task, the agent interacts with the environment, whose next state (or the probability of it) is fully dependent on the current state and action. Specifically, the prompt itself already includes the task description and interaction history, which contains all decision-relevant information needed; an expert can act optimally from this single prompt, making the process effectively Markovian. Also, this is how recurrent RL, world-model planning (Dreamer[1], MuZero[2]), WebDreamer-style LLM agents[3] operate.
>
> #### 3.2 Minecraft can be modeled as an MDP.
> Several prior works explicitly formulate Minecraft tasks as MDPs once the state is appropriately defined, for example, Scheller et al. (2020) reduce MineRL ObtainDiamond from a POMDP to a “standard MDP (S, A, P, R)” via LSTM memory[4]; Aluru et al. (2015) use an Object-Oriented MDP representation[5]. Frazier & Riedl (2019)[6] train deep reinforcement learning agents in Minecraft and apply interactive human action advice in an MDP-style RL loop, showing that Minecraft behaviour problems are amenable to the MDP formalism. This shows such environments (Minecraft) can be modeled as MDPs without issue back to 10 years ago.
> #### 3.3. MDP is the common choice in RL and modern LLM-agent work.
> MDP is the keystone of many RL formulations. In fact, almost all RL methods including behavior Cloning (BC), offline RL, valued-based method and policy based method (PPO) all start from the MDP formulation and its extensions (e.g. POMDP and hierarchical MDP).
> In addition, recent LLM-agent papers (e.g., [7-11]) all adopt the MDP formalism, where the “state” is the text observation plus interaction history. This setting works cleanly for our eight evaluated environments and is the de-facto standard for current LLM-based agents.
>
> #### 3.4 Clarification regarding [Tanskanen et al]
> We appreciate that the reviewer raised a related work regarding minecraft. However, from this paper itself the authors are still following an MDP setting (Section 3). We would appreciate it if the reviewer *could explicitly clarify the reason why MDP assumption is not suitable for our RL task*.
>
> [Sutton and Barto] Reinforcement Learning: An Introduction, MIT Press 1998.
>
> [1] Hafner et al., Dream to Control: Learning Behaviors by Latent Imagination, ICLR 2020.
>
> [2] Schrittwieser et al., Mastering Atari, Go, Chess and Shogi by Planning with a Learned Model, Nature, 2020.
>
> [3] Gu et al., Is Your LLM Secretly a World Model of the Internet? Model-Based Planning for Web Agents, TMLR 2025.
>
> [4] Scheller et al., Sample Efficient Reinforcement Learning through Learning from Demonstrations in Minecraft, NeurIPS 2019.
>
> [5] Aluru et al., Minecraft as an Experimental World for AI in Robotics, AAAI 2015.
>
> [6] Frazier et al., Improving Deep Reinforcement Learning in Minecraft with Action Advice, AAAI 2019.
>
> [7] Feng et al., Group-in-Group Policy Optimization for LLM Agent Training, NeurIPS 2025.
>
> [8] Shi et al., MobileGUI-RL: Advancing Mobile GUI Agent through Reinforcement Learning in Online Environment, 2025.
>
> [9] Zhang et al., RLVMR: Reinforcement Learning with Verifiable Meta-Reasoning Rewards for Robust Long-Horizon Agents, 2025.
>
> [10] Wang et al., RAGEN: Understanding Self-Evolution in LLM Agents via Multi-Turn Reinforcement Learning, 2025.
>
> [11] Zhang et al., AGENTRL: Scaling Agentic Reinforcement Learning with a Multi-Turn, Multi-Task Framework, 2025.
>
>
> ### 4. Regularized BC
>
> We agree that advanced IL methods, including regularized BC, can be useful in general. However, this line of work is orthogonal to our contribution. Our goal is not to improve IL itself. We study a **complementary** learning paradigm that improves LLM-based policies even when IL or RL pipelines are limited, unavailable, or costly to run. In this sense, our method can support or enhance advanced IL approaches rather than replace them.
> Exploring how our approach interacts with stronger IL variants is valuable future work, but it is outside the scope of this paper. Our contribution is to broaden the design space of policy learning for language agents, not to propose an alternative to regularized BC.

---

> > ### Author Response · Authors · 2025-11-20
> > **Response to Reviewer MfPf (3/3)**
> >
> > ### 5. Adversarial IL
> > Adversarial imitation learning (AIL) introduces substantial training overhead. AIL requires maintaining two models simultaneously: a policy model and a discriminator that judges expert versus agent behavior. The training loop must alternate between these two models, e.g., generating trajectories, evaluating them with the discriminator, and updating both models. For LLM-based agents, this online dual-model optimization makes training significantly slower and less stable in practice.
> > Also, in preliminary experiments on ALFWorld, this did not offer clear benefits. Since expert demonstrations in these environments are already strong, the discriminator signal was reduced to “match the expert action,” which produced performance similar to standard behavior cloning.
> > Our methods target a different objective. Implicit world modeling and self-reflection supply additional supervision without maintaining a second model or running an adversarial loop. This single-model design is simpler, more stable, and provides information (future-state prediction and reasoning about action quality) that AIL does not provide.
> > AIL is orthogonal to our method. It can be combined with it, but it requires a different and more expensive training setup than the efficient single-model paradigm we study.
> >
> > ### Q. World Model for Trajectory Prediction
> > In our method, the “world model” is **implicit**: we **do not** use it to generate any steps during training or inference. All future states come directly from the environment. The next-state prediction objective simply helps the policy internalize how the environment changes after an action. Thus, our world-modeling stage is intentionally designed for one-step prediction, not for simulating longer trajectories.
> > We have also experimented with multi-step predictions in pilot studies, and the policy indeed benefits from the additional environmental knowledge. However, learning stable long-horizon transition dynamics in complex web and tool-use environments remains challenging and often unstable [1,2]. Extending implicit world modeling to reliable multi-step rollouts is an interesting direction for future work, but it is outside the scope of this paper.
> >
> > [1] Gu et al., Is Your LLM Secretly a World Model of the Internet? Model-Based Planning for Web Agents, TMLR 2025.
> >
> > [2] Chae et al., Web Agents with World Models: Learning and Leveraging Environment Dynamics in Web Navigation, ICLR 2025.

---

### Author Response · Authors · 2025-11-20
**General Response (1/3)**

We thank the AC and all reviewers for their time. Below we provide general clarifications related to common questions raised across reviews and additional experiments we did.
Between submission and review release, we also conducted additional experiments that we believe could further strengthen our results, such as incorporating the Qwen2.5 model family and adding clearer floating tables/figures. We will integrate these updates in the next revision.


### 1. Clarification on the key differences between previous agents and language agents
*(All four reviews)*
Much of the related work that our reviewers reference (e.g., classical world-model RL, adversarial imitation learning, regularized BC) was developed for small neural policies acting in low-dimensional state and action spaces with well-defined scalar rewards. In contrast, our setting is that of LLM-based agents/policies, which differ along several crucial axes:

- **Representation and I/O**[1,2,3]. States, actions, and intermediate reasoning are all expressed in natural language (tool calls and ‘turn left 60 degree’ actions), rather than compact vectors. The same backbone can produce next states, actions, and rationales in a unified text format. Because of this flexibility of language and powerful foundation of LLM, language agents can (i) produce actions, (ii) predict next states in language, and (iii) explain why expert actions are preferred without introducing new heads or parameters.

- **Environment regime**[4,5,6]. Language agents are developed for challenging, real-world, and interactive environments (browsers, tool APIs, simulators), many of them do not provide reliable reward functions. This is the opposite of many classical RL benchmarks where the reward is often known and well-defined. We explained this in the abstract and introduction and our “early experience” stage is therefore designed as a reward-free, BC-compatible augmentation that fits into existing LLM training infrastructure.

- **Supervision and objectives**[7,8]. Training is dominated by next-token prediction over heterogeneous traces (instructions, tools, environment observations, chain-of-thought rationales), not by direct action loss or temporal-difference updates. On environments mentioned above (web, complex tool use), reward signals are sparse, non-verifiable, or even unavailable, which makes standard RL or adversarial IL pipelines difficult to deploy.

This is why we view our two methods, implicit world modeling and self-reflection, here not as a direct re-use of prior RL ideas, but as LLM-specific instantiations that exploit the language interface.
To the best of knowledge, we haven’t noticed any work using implicit world modeling or self-reflection for language agents yet.
Our experiments across eight language-agent benchmarks and multiple model families indicate that this reward-free early experience stage is a practical and scalable way to improve LLM agents in precisely the environments where classical reward-driven agents are hardest to apply.

[1] Yao et al., ReAct: Synergizing Reasoning and Acting in Language Models, ICLR 2023.

[2] Sumers et al., Cognitive Architectures for Language Agents, TMLR 2024.

[3] Shinn et al., Reflexion: Language Agents with Verbal Reinforcement Learning, NeurIPS 2023.

[4] Patil et al., The Berkeley Function Calling Leaderboard (BFCL): From Tool Use to Agentic Evaluation of Large Language Models, ICML 2025.

[5] Liu et al., AgentBench: Evaluating LLMs as Agents, ICLR 2024.

[6] Xie et al., TravelPlanner: A Benchmark for Real-World Planning with Language Agents, ICML 2024.

[7] Deng et al., Mind2Web: Towards a Generalist Agent for the Web, NeurIPS 2023.

[8] Pahuja et al., Explorer: Scaling Exploration-drivenWeb Trajectory Synthesis for Multimodal Web Agents, ACL 2025.

---

> ### Author Response · Authors · 2025-11-20
> **General Response (2/3)**
>
> ### 2. Table: Eight Benchmarks Used Across Three Domains
> *(Q6 from Reviewer LDNN and W1/Q7 from Reviewer b21C)*
> For clarity, we summarize the environments used across our three domains and include the number of expert trajectories and state–action pairs. In the current version we mentioned these environments in detail in Appendix E.2 due to space limits. We will move this table to the main paper if space permits. Together with the results in the main text, this illustrates that our method works across environments with widely varying amounts of expert data.
>
>
> #### Benchmarks Used Across Three Domains
> **“# Traj.”** = number of expert trajectories collected/used.
> **“# D_expert”** = number of state–action (SA) pairs for imitation learning.
>
> | Environment | Description | # Traj. | # D_expert |
> |------------|-------------|---------|-------------|
> | **_MISC (Embodied / Scientific Simulation / Travel Planning)_** |  |  |  |
> | **ALFWorld** (Shridhar et al., 2021) | Embodied instruction-following in simulated households with textual observations and symbolic actions. Following Feng et al., 2025. | 3,553 | 21,031 |
> | **ScienceWorld** (Wang et al., 2022) | Interactive science-lab simulator rendered in natural language; multi-step tool-use experiments. Gym wrapper implemented. | 1,000 | 14,506 |
> | **TravelPlanner** (Xie et al., 2024) | Long-horizon itinerary planning using various tools; sole-planning mode with gym wrapper. | 45 | 1,395 |
> | **_Multi-Turn Tool Use_** |  |  |  |
> | **BFCLv3** (Patil et al., 2025) | Multi-turn tool-use tasks from Berkeley Function Call Leaderboard v3 using Python-based API tools. | 125 | 1,264 |
> | **Tau-Bench (Retail)** (Yao et al., 2025) | Customer-service scenarios requiring multi-turn API use while following policy documents. | 452 | 5,239 |
> | **SearchQA** (Jin et al., 2025) | Multi-hop QA using search tools; Musique (ID), and HotpotQA/2WikiMultiHopQA/Bamboogle (OOD). | 2,082 | 7,691 |
> | **_Web Navigation_** |  |  |  |
> | **WebShop** (Yao et al., 2022) | Product search and selection in simulated e-commerce sites. Following Feng et al., 2025. | 1,571 | 15,464 |
> | **WebArena-Lite** (Zhou et al., 2023; Liu et al., 2025) | Web navigation across domains (e-commerce, forums, CMS) with accessibility-tree observations (Koh et al., 2024). | 554 | 7,044 |
>
>
>
>
>
>
> ### 3. More Results with Llama and Qwen Models Across Eight Benchmarks
> *(Q8 from Reviewer b21C and W1 from Reviewer LDNN)*
>
> We include updated results with the Qwen2.5 model family to show that our method generalizes across both models and environments.
>
>
> #### Table: Results Across Eight Benchmarks (Success Rates %)
> All values are success rates (%) unless otherwise noted.
> “IWM” = Implicit World Modeling, “SR” = Self-Reflection.
> Improvements over imitation learning are shown in parentheses.
>
> | Benchmark | Model | Prompt | Imitation Learning | Ours-IWM | Ours-SR |
> |-----------|--------|--------|---------------------|-----------|-----------|
> | **_Embodied & Scientific Simulation & Travel Planning_** |  |  |  |  |  |
> | **ALFWorld** | Llama-3.2-3B | 8.6 | 78.1 | 83.6 (+5.5) | **85.9 (+7.8)** |
> |  | Qwen-2.5-7B | 20.3 | 78.1 | **82.8 (+4.7)** | 82.0 (+3.9) |
> |  | Llama-3.1-8B | 25.0 | 80.5 | **85.9 (+5.4)** | 85.2 (+4.7) |
> | **ScienceWorld** | Llama-3.2-3B | 2.3 | 51.6 | 55.5 (+3.9) | **56.2 (+4.6)** |
> |  | Qwen-2.5-7B | 3.9 | 53.9 | **59.4 (+5.5)** | 57.8 (+3.9) |
> |  | Llama-3.1-8B | 3.1 | 54.7 | 57.0 (+2.3) | **68.0 (+13.3)** |
> | **TravelPlanner** | Llama-3.2-3B | 0.0 | 19.4 | 28.3 (+8.9) | **32.2 (+12.8)** |
> |  | Qwen-2.5-7B | 0.0 | 16.7 | 22.2 (+5.5) | **31.7 (+15.0)** |
> |  | Llama-3.1-8B | 0.0 | 17.2 | 25.0 (+7.8) | **32.2 (+15.0)** |
> | **_Multi-Turn Tool Use_** |  |  |  |  |  |
> | **BFCLv3** | Llama-3.2-3B | 1.3 | 21.3 | 25.3 (+4.0) | **29.3 (+8.0)** |
> |  | Qwen-2.5-7B | 10.6 | 26.7 | 29.3 (+2.6) | **32.0 (+5.3)** |
> |  | Llama-3.1-8B | 6.7 | 16.0 | **20.0 (+4.0)** | **20.0 (+4.0)** |
> | **Tau-Bench** | Llama-3.2-3B | 5.2 | 24.3 | 26.1 (+1.8) | **28.7 (+4.4)** |
> |  | Qwen-2.5-7B | 20.0 | 33.9 | 38.7 (+4.8) | **39.5 (+5.6)** |
> |  | Llama-3.1-8B | 6.0 | 35.9 | 40.8 (+4.9) | **41.7 (+5.8)** |
> | **SearchQA (F1)** | Llama-3.2-3B | 13.3 | 38.0 | **39.0 (+1.0)** | 38.6 (+0.6) |
> |  | Qwen-2.5-7B | 19.3 | 39.9 | 40.8 (+0.9) | **42.0 (+2.1)** |
> |  | Llama-3.1-8B | 21.0 | 41.0 | **44.3 (+3.3)** | 41.8 (+0.8) |
> | **_Web Navigation_** |  |  |  |  |  |
> | **WebShop** | Llama-3.2-3B | 0.0 | 41.8 | **60.2 (+18.4)** | 52.7 (+10.9) |
> |  | Qwen-2.5-7B | 0.8 | 51.6 | 56.2 (+4.6) | **62.2 (+10.6)** |
> |  | Llama-3.1-8B | 0.0 | 47.3 | **58.6 (+11.3)** | 58.2 (+10.9) |
> | **WebArena-Lite** | Llama-3.2-3B | 1.2 | 6.1 | **8.5 (+2.4)** | 7.3 (+1.2) |
> |  | Qwen-2.5-7B | 1.8 | 4.2 | **7.3 (+3.1)** | 6.1 (+1.9) |
> |  | Llama-3.1-8B | 0.6 | 4.9 | **8.5 (+3.6)** | **8.5 (+3.6)** |

---

> ### Author Response · Authors · 2025-11-20
> **General Response (3/3)**
>
> ### 4. Model Scaling on WebArena-Lite
> *(Q4 from Reviewer b21C)*
> We further evaluate our method with larger open-source models (up to 70B–72B scale). Early Experience continues to improve over imitation learning across model sizes different model sizes and families.
>
>
>
> ### Table: Model Scaling on WebArena-Lite (Q4 from Reviewer b21C)
>
> | Type | Method | Reddit | Gitlab | CMS | Map | OSS | Avg. SR |
> |------|--------|--------|--------|-----|-----|-----|---------|
> | **_Base: Closed-Source Model_** |  |  |  |  |  |  |  |
> | Prompting | GPT-4-Turbo | 10.5 | 16.7 | 14.3 | 36.7 | 13.3 | 17.6 |
> | Prompting | GPT-4o | 10.5 | 10.0 | 20.0 | 20.0 | 11.1 | 13.9 |
> | **_Base: Llama-3.2-3B-Instruct_** |  |  |  |  |  |  |  |
> | Prompting | – | 0.0 | 0.0 | 0.0 | 1.7 | 3.2 | 1.2 |
> | Imitation Learning | Behavior Cloning | 0.0 | 7.7 | 10.3 | 0.0 | 3.2 | 6.1 |
> | Early Experience | Implicit World Modeling | 0.0 | 15.4 | 2.4 | 13.8 | 9.7 | 8.5 |
> | Early Experience | Self-Reflection | 11.1 | 15.4 | 11.9 | 3.5 | 6.5 | 7.3 |
> | **_Base: Qwen-2.5-7B-Instruct_** |  |  |  |  |  |  |  |
> | Prompting | – | 0.0 | 7.7 | 2.4 | 1.8 | 0.0 | 1.8 |
> | Imitation Learning | Behavior Cloning | 12.5 | 0.0 | 7.4 | 4.0 | 9.1 | 4.2 |
> | Early Experience | Implicit World Modeling | 0.0 | 15.4 | 7.1 | 8.6 | 6.5 | 7.3 |
> | Early Experience | Self-Reflection | 0.0 | 7.7 | 11.9 | 5.2 | 6.5 | 6.1 |
> | **_Base: Llama-3.1-8B-Instruct_** |  |  |  |  |  |  |  |
> | Prompting | – | 0.0 | 0.0 | 7.1 | 0.0 | 0.0 | 0.6 |
> | Imitation Learning | Behavior Cloning | 0.0 | 0.0 | 0.0 | 11.9 | 8.0 | 4.9 |
> | Early Experience | Implicit World Modeling | 11.1 | 0.0 | 7.3 | 8.6 | 16.1 | 8.5 |
> | Early Experience | Self-Reflection | 0.0 | 15.4 | 11.9 | 10.3 | 3.2 | 8.5 |
> | **_Base: Llama-3.3-70B-Instruct_** |  |  |  |  |  |  |  |
> | Prompting | – | 11.1 | 15.4 | 17.1 | 7.1 | 3.2 | 9.1 |
> | Imitation Learning | Behavior Cloning | 0.0 | 8.3 | 14.3 | 17.2 | 16.1 | 13.3 |
> | Early Experience | Implicit World Modeling | 8.3 | 16.7 | 16.7 | 19.0 | 19.4 | 16.4 |
> | Early Experience | Self-Reflection | 0.0 | 16.7 | 23.8 | 14.0 | 19.4 | 15.2 |
> | **_Base: Qwen-2.5-72B-Instruct_** |  |  |  |  |  |  |  |
> | Prompting | – | 12.5 | 15.4 | 12.2 | 6.9 | 6.7 | 8.5 |
> | Imitation Learning | Behavior Cloning | 0.0 | 7.7 | 19.0 | 12.3 | 16.7 | 12.7 |
> | Early Experience | Implicit World Modeling | 25.0 | 15.4 | 26.2 | 12.3 | 19.4 | 17.6 |
> | Early Experience | Self-Reflection | 0.0 | 15.4 | 23.8 | 14.0 | 20.0 | 15.8 |

---

### Meta-Review · Area_Chair_g7Ek · 2025-12-23

**Summary:**

Reviewers generally agree that the paper proposes a reasonable and practical reward-free policy improvement framework for LLM agents, introducing implicit world modeling and self-reflection as a middle ground between imitation learning and reinforcement learning. The approach is well-motivated for environments without verifiable rewards and shows consistent empirical gains across diverse benchmarks and model scales. However, the decision is mainly influenced by concerns about limited novelty, the incremental nature of the contribution.

**Reviewer Concerns:**

**Addressed by the rebuttal:**

Clarified the novelty in the LLM-agent setting compared to prior world-model and imitation-learning methods.

Justified the MDP formulation and environment observability assumptions.

Added comparisons with DPO and results across different model families and scales.

Improved clarity on training setup and self-reflection design.

**Still outstanding:**

Novelty is still viewed by some reviewers as limited.

Long-horizon reasoning and statistical rigor remain insufficiently explored.

The method relies on accessible and resettable environments.

**Reviewer Scores:**

Reviewer MfPf: Likely unchanged or slightly improved (still negative on novelty).

Reviewer sfTJ: Likely unchanged, leaning borderline.

Reviewer LDNN: Likely improved.

Reviewer b21C: Likely slightly improved toward borderline accept.

---

### Decision · Program_Chairs · 2026-01-26

Reject